# Exogenous 2-keto-L-gulonic Acid Supplementation as a Novel Approach to Enhancing L-ascorbic Acid Biosynthesis in Zebrafish (*Danio rerio*)

**DOI:** 10.3390/ani13152502

**Published:** 2023-08-03

**Authors:** Meijun Shi, Mingfu Gao, Hao Sun, Weichao Yang, Hongxia Zhao, Lixin Zhang, Hui Xu

**Affiliations:** 1CAS Key Laboratory of Forest Ecology and Management, Institute of Applied Ecology, Chinese Academy of Sciences, Shenyang 110016, China; 2University of Chinese Academy of Sciences, Beijing 100049, China; 3Key Laboratory of Pollution Ecology and Environmental Engineering, Institute of Applied Ecology, Chinese Academy of Sciences, Shenyang 110016, China; 4Modern Agricultural Science and Technology Innovation Center of Kuqa City, Kuqa 842000, China; 5State Key Laboratory of Bioreactor Engineering and School of Biotechnology, East China University of Science and Technology, Shanghai 200237, China

**Keywords:** 2-keto-L-gulonic acid, zebrafish (*Danio rerio*), L-ascorbic acid biosynthesis, growth performance, feed utilization, whole-body chemical composition

## Abstract

**Simple Summary:**

L-ascorbic acid plays a significant role in maintaining the physiological functions and survival of aquatic animals. This is the first report on elevating L-ascorbic acid accumulation in aquatic animals through exogenous 2-keto-L-gulonic acid supplementation. This study found the positive effects of exogenous 2-keto-L-gulonic acid supplementation on L-ascorbic acid metabolism and growth in zebrafish (*Danio rerio*), such as the enhanced L-ascorbic acid content, growth performance and feed utilization, crude protein, and crude lipid content. Our findings suggest that 2-keto-L-gulonic acid has the potential to act as a precursor of L-ascorbic acid, and supplementing with 2-keto-L-gulonic acid could be a novel strategy for enhancing L-ascorbic acid levels in aquatic animals.

**Abstract:**

L-ascorbic acid (ASA) is a micronutrient that is essential for reproduction, growth, and immunity in animals. Due to the loss of enzyme L-gulono-1,4-lactone oxidase (GLO), most aquatic animals lack the capacity for ASA biosynthesis and therefore require supplementation with exogenous ASA. Recent studies have shown that 2-keto-L-gulonic acid (2KGA), a novel potential precursor of ASA, can enhance plant growth and improve stress resistance by promoting the synthesis and accumulation of ASA. Our hypothesis is that 2-keto-L-gulonic acid (2KGA) plays a similar role in aquatic animals. To investigate this, we conducted an in vivo trial to examine the effects of exogenous 2KGA supplementation on ASA metabolism and growth of zebrafish (*Danio rerio*). Zebrafish were categorized into groups based on their dietary intake, including a basal diet (CK group), a basal diet supplemented with 800 mg/kg ASA (ASA group), and 800 mg/kg 2KGA-Na (2KGA group) for a duration of three weeks. The results demonstrated a significant increase in ASA content in zebrafish treated with 2KGA (34.82% increase, *p* < 0.05) compared to the CK group, reaching a consistent level with the ASA group (39.61% increase, *p* < 0.05). Furthermore, the supplementation of 2KGA significantly improved growth parameters relevant to zebrafish (specific growth rate increased by 129.04%, *p* < 0.05) and enhanced feed utilization (feed intake increased by 15.65%, *p* < 0.05). Positive correlations were observed between growth parameters, feed utilization, whole-body chemical composition, and ASA content. Our findings suggest that supplementation with exogenous 2KGA can serve as a novel approach for elevating ASA synthesis in aquatic animals, and further investigation of its underlying mechanism is required.

## 1. Introduction

L-ascorbic acid (ASA, Vitamin C) is an essential nutrient for the normal physiological function of most aquatic animals [1]. Many studies have shown the advantageous effects of ASA on growth, immunological parameters, and enhanced resistance to stress and diseases [2,3,4]. ASA is also extensively utilized in the aquaculture industry as an important nutrient and immune modulator for aquatic animals [5,6]. However, many aquatic animals have a limited ability to synthesize ASA due to the absence of L-gulono-1,4-lactone oxidase (GLO), which is responsible for ASA biosynthesis [7]. Therefore, it is crucial to ensure sufficient intake of ASA through feed supplementation for maintaining the health and optimizing the growth performance of aquatic animals.

Supplemental ASA (25–2000 mg/kg) has been reported to have a positive impact on growth performance and immunological responses in various aquatic animals, thereby enhancing their resistance to stressors [8,9,10]. However, ASA is unstable, its oxidation is accelerated in alkaline environments, and the rate of its loss is influenced by factors such as temperature, oxygen, light, and pH [11,12]. Even when provided with 4–5 times the recommended dosage of ASA, its efficacy still cannot be guaranteed [13]. Currently, the market applications of encapsulated ASA or its phosphate derivatives are costly, and their biosafety remains to be evaluated [14,15,16,17]. Therefore, it is important to develop novel technologies to increase the content of ASA in aquatic animals and improve its stability and bioavailability.

2-keto-L-gulonic acid (2KGA), an exclusive precursor of ASA in industrial production, is mainly produced via a two-step microbial fermentation process involving *Gluconobacter oxydans*, *Ketogulonicigenium vulgare*, and *Bacillus megaterium* [18,19,20]. However, during the crystallization of 2KGA from fermented solution, a large amount of residue after evaporation (RAE) is discharged, mainly consisting of 2KGA (approximately 25%), formic acid, oxalic acid, proteins, nucleic acid, and water [21]. In our previous study on the resourceful utilization of RAE, it was found that RAE effectively promotes the growth of *Brassica campestris* L., with a 23.9% increase in biomass and a 188.94% increase in ASA content [22]. Subsequently, Gao et al. [23] reported that exogenous 2KGA can increase ASA accumulation and the expression of enzymes related to ASA synthesis and stress resistance, thereby alleviating the inhibitory effects of salt stress on plant growth. Recently, we have demonstrated through stable isotope tracer experiments that 2KGA serves as a precursor for the synthesis of ASA in plants and also exerts a facilitative effect on plant growth and development (unpublished data). Thus, we conclusively confirmed that ASA synthesis through the utilization of exogenous 2KGA in plants is a novel phenomenon. As an essential vitamin, the biosynthetic pathway of ASA in animals partially intersects with that of plants and some of the synthetic genes are homologous [7]. In most animals, the final step of ASA biosynthesis is the GLO-catalyzed conversion of L-gulono-1,4-lactone to ASA, which is the same as in the L-gulose pathway of plants (Figure 1) [7,24,25,26,27]. Hence, it would be a captivating research topic to explore whether 2KGA exerts an impact on the levels of ASA accumulation in animals.

Up to now, there is no report on whether exogenous 2KGA affects the ASA synthesis process in animals, including aquatic animals. Based on the positive role of 2KGA in the ASA synthesis of plants and the significance of ASA for aquatic animal health, we proposed the following hypothesis: exogenous 2KGA is involved in the process of ASA biosynthesis in aquatic animals, thereby enhancing their growth status and resistance to environmental stresses. Zebrafish (*Danio rerio*) are an excellent vertebrate model in various fields of aquaculture research [28], and are currently being extensively utilized for fish nutritional evaluation and metabolic mechanism studies [29,30]. Thus, in this study, we investigated the potential of exogenous 2KGA supplementation to enhance animal growth and increase ASA accumulation in zebrafish as a model organism.

## 2. Materials and Methods

### 2.1. Animals and Experimental Conditions

Zebrafish (*Danio rerio*) were supplied from the China Zebrafish Resource Center, CZRC (Wuhan, China). The zebrafish were, on average, ~3.4 cm in body length and 6 months of age. They were bred at a light: dark photoperiod (14-h:10-h) in 10 L cylindrical FRP tanks with 8 L water in each. The water source was aerated tap water treated with sodium sulfate, with a daily water change rate of 50% and continuous aeration. Temperature, pH, and dissolved oxygen were maintained at 25 °C, 7.5–7.9, and 5–8 mg/L, respectively. Prior to the experiment, the zebrafish were acclimatized to the culture environment for three weeks and fed the basal diet. The experimental conditions were standardized for zebrafish throughout the experimental period.

### 2.2. Experimental Diets

A basal diet was formulated using soybean meal, rapeseed meal, and fishmeal as the main protein sources, with soybean oil, fish oil, and soybean lecithin as the main lipid sources. Wheat middling and wheat gluten were used as the main sugar sources. 2KGA-Na (800 mg/kg) was supplemented to the basal diet for a treatment group, while ASA (800 mg/kg) was added to the basal diet for a positive control group (2KGA-Na (>99.5%) and ASA (>99.5%) were supplied by Northeast Pharmaceutical Group Co., Ltd., Shenyang, China). The negative control group was administered the basal diet only. The formulation and proximate composition of the basal and experimental diets are shown in Appendix A. The main ingredients of the basal diet were supplied by Tech-bank Food Co., Ltd., Ningbo, China. The ingredients were pulverized to a particle size that could pass through an 80 μm sieve, thoroughly mixed. Afterwards, the oil and distilled water were added and mixed, and then extruded into 1.0–1.5 mm pellets using a pelletizer. Finally, they were dried at 60 °C for 12 h before being stored at −20 °C until application.

### 2.3. Experimental Design

After three weeks of acclimation, a total of 90 healthy and uniformly sized zebrafish were distributed into three groups: CK group (Negative control group), 2KGA group (Treatment group), and ASA group (Positive control group). Each experimental group consisted of three replicates (tanks), with ten zebrafish in each replicate. Each FRP tank was filled with 8 L of water. The initial weights of the three groups of zebrafish were not significantly different (Table 1). Each experimental group was fed with their corresponding experimental feed, and all experimental zebrafish were fed to apparent satiation twice daily at 10:00 and 18:00 for three weeks. All the zebrafish were maintained under the aforementioned acclimatized conditions. We conducted regular monitoring of the physicochemical parameters of the aquarium water. The animal study was reviewed and approved by the Animal Ethics Committee of the Institute of Applied Ecology, Chinese Academy of Science.

### 2.4. Extraction and Determination of ASA

At the end of the rearing trial, all zebrafish were fasted for 24 h and anesthetized with MS-222 (0.1 g/L, Sigma-Aldrich, Shanghai, China). Five zebrafish were randomly selected from each replicate for weighing in order to calculate growth parameter indices and feed utilization. Subsequently, ASA measurements were taken. The samples of zebrafish were extracted using a 1% metaphosphoric acid solution for ASA concentration measurement. After centrifugation at 6000 r/min for 10 min, the supernatant was collected for ASA content determination by means of high-performance liquid chromatography (HPLC). The detection conditions were as follows: mobile phase of 5% acetonitrile and 95% NaH_2_PO_4_ solution (20 mM, adjusted pH to 2.8 ± 0.1 with 10% HCL solution) at a flow rate of 1 mL/min; temperature, 40 °C; detection wavelength of 243 nm [23,31].

### 2.5. Analysis of Growth Performance and Feed Utilization Parameters

The growth indicators and feed utilization parameters were calculated based on the relevant indexes as follows:-Weight gain rate (WGR, %) = 100 × [final mean weight (g) − initial mean weight (g)]/initial weight (g).-Specific growth rate (SGR, %) = 100 × [Ln final mean weight (g) − Ln initial mean weight (g)/feeding days].-Feed conversion ratio (FCR, %) = 100 × Feed consumed (g)/weight gain (g).-Protein efficiency ratio (PER, g/g) = experimental animal gain (g)/protein intake (g).-Condition factor (CF, g/cm^3^) = 100 × body weight (g)/body length (cm^3^).-Survival rate (%) = 100 × final number of animals/initial number of animals.-Hepato-somatic index (HSI, %) = 100 × liver weight (g)/body weight (g).-Gonado-somatic index (GSI, %) = 100 × Gonad weight (g)/body weight (g).

Feed intake (FI, g) was the amount of feed provided during the experiment after excluding the uneaten feed by collecting and drying the uneaten feed half an hour after each meal [32,33,34].

### 2.6. Whole-Body Proximate Chemical Composition

Three weeks later, three zebrafish were randomly selected from each replicate for measurement of their whole-body proximate chemical composition. Moisture, crude protein, crude lipid, and crude ash content were measured according to AOAC procedures [35]. Moisture was determined by placing the zebrafish in a 105 °C oven until a constant weight was reached, and then calculating the difference in weight. Crude protein content was measured by the Kjeltec™ 2300 analyzer unit (FOSS Analytical, Slangerupgade, DK-3400 Hilleroed, Denmark). Crude lipid determination was performed using a soxhlet analyzer (VELP Scientifica, SER 148, Usmate Velate, Italy), and crude ash determination was conducted by weighing the crude ash residue after burning the fish sample in a muffle furnace at 550 °C for 12 h.

### 2.7. Data Analysis

All the statistical analyses were performed using GraphPad Prism (Version Prism 8) and R software (version 4.0.2, http://www.r-project.org/, accessed on 30 January 2022). Data were expressed as mean ± standard error (SE). One-way analysis of variance (ANOVA) and Tukey’s multiple comparison tests were used to identify significant variations. Spearman’s correlation analysis was used to calculate the correlation coefficients between ASA, physiological indicators, and feed utilization. *p* < 0.05 was considered to indicate a statistically significant difference.

## 3. Results

### 3.1. Effect of Exogenous 2KGA on ASA Levels in Zebrafish

The results indicated that 2KGA addition significantly elevated the ASA levels of zebrafish on the 21st day (Figure 2). Compared to the CK group (51.87 ± 14.43 μg/g), the ASA concentrations of zebrafish in the 2KGA group (69.93 ± 18.36 μg/g) and the ASA group (72.41 ± 23.61 μg/g) were increased by 34.82 and 39.61% (Figure 2a), respectively. An increase in endogenous ASA levels of 18.06 μg/g and 20.54 μg/g was observed in 2KGA and ASA groups, respectively, compared to the control group (Figure 2a) (*p* < 0.05).

Meanwhile, the total ASA contents of zebrafish increased by 43.68% and 47.61% in the 2KGA group (40.84 ± 2.09 μg/individual) and ASA group (41.96 ± 1.68 μg/individual), respectively, compared to the CK group (28.42 ± 2.08 μg/individual) (Figure 2b) (*p* < 0.05). The disparity in ASA production between the 2KGA and CK groups of zebrafish may provide us with a value for the ASA synthesized exclusively from the 2KGA, which is 12.42 μg/individual (Figure 2b).

However, no significant difference was observed in ASA concentrations and the total ASA contents of zebrafish between the 2KGA and the ASA groups (Figure 2) (*p* > 0.05).

### 3.2. Growth Performance Parameters

After three weeks of feeding, the weight gain rate (WGR) and specific growth rate (SGR) were significantly higher in the 2KGA and ASA groups of zebrafish than those of the CK group (Figure 3). The WGR of zebrafish treated with 2KGA and ASA increased by 148.67 and 183.52%, respectively, compared to the CK group (Figure 3a) (*p* < 0.05), whereas SGR increased by 129.04% and 167.77%, respectively (Figure 3b). There were no significant differences in the values of condition factor (CF), Hepato-somatic index (HSI), and Gonado-somatic index (GSI) among any of the groups (Figure 3c–e) (*p* > 0.05). Compared to the CK group (90%), the survival rates of the 2KGA and the ASA groups were 93% and 90%, respectively (Figure 3f) (*p* > 0.05).

The moisture of zebrafish did not show significant differences among treatments (Figure 4a) (*p* > 0.05). The 2KGA and the ASA treatment significantly boosted the levels of crude protein and crude lipid in zebrafish. The crude protein contents of zebrafish treated with 2KGA and ASA increased by 5.75 and 8.52%, respectively, compared to the CK group (Figure 4b) (*p* < 0.05). Similarly, the crude lipid contents increased by 34.93 and 22.16%, respectively (Figure 4c) (*p* < 0.05). The crude ash content levels decreased in zebrafish reared under 2KGA treatments compared to the CK group (15.76% decrease) (Figure 4d) (*p* < 0.05). The crude ash content under ASA treatments also decreased by 14.46% (Figure 4d) (*p* < 0.05).

### 3.3. Feed Utilization Parameters

The inclusion of 2KGA and ASA in the diet resulted in increased feed intake (FI) and protein efficiency ratio (PER) for zebrafish compared to the control group (CK) (Figure 5). The feed conversion ratio (FCR) of zebrafish decreased when 2KGA and ASA were added (a 61.99 and 64.84% decrease, respectively) (Figure 5a). The PER increased by 130.76 and 143.91%, respectively, in zebrafish (Figure 5b) (*p* < 0.05). Meanwhile, the FI of zebrafish treated with 2KGA and ASA also increased by 15.65 and 9.72%, respectively, compared to the CK group (Figure 5c). A low feed conversion ratio (FCR) is the primary indicator of efficient aquaculture because it suggests superior feed utilization efficiency for the 2KGA group in zebrafish.

### 3.4. Analysis of Correlations between ASA Content, Growth Parameters, Feed Utilization

In zebrafish under 2KGA treatment, variation of ASA levels was positively correlated significantly with the specific growth rate, weight gain rate, protein efficiency ratio, feed intake, crude protein, and crude lipid levels (*r* > 0.65, *p* < 0.05) and negatively correlated significantly with feed conversion ratio and crude ash level (Figure 6) (*r* < −0.85, *p* < 0.05). It implied that 2KGA may play a role in regulating the health of aquatic animals by enhancing ASA levels.

## 4. Discussion

In recent studies, we have demonstrated that exogenous 2-keto-L-gulonic acid (2KGA) serves as a direct substrate for the synthesis of L-ascorbic acid (ASA) in plants, thereby effectively promoting ASA accumulation. In addition to *Arabidopsis thaliana*, a significant increase in ASA levels was observed across several field crops, including *Brassica campestris* ssp. *chinensis*, *Spinacia oleracea*, *Piper nigrum*, and *Brassica oleracea* upon treatment with 2KGA. Therefore, the universality of 2KGA in enhancing plant ASA levels has been further validated, and its positive effects on plant growth and resistance to stress have been further proven [23]. However, until now, there have been no reports regarding the impact of exogenous 2KGA supplementation on ASA synthesis in animals. In this study, 800 mg/kg of 2KGA-Na significantly increased the ASA concentration in zebrafish by 1.35-fol, compared to the control group. To our best knowledge, this is the first report to demonstrate an increase in ASA accumulation in aquatic animals through exogenous 2KGA supplementation. 

The improvements of ASA for 2KGA supplementation are almost equivalent to those for direct ASA supplementation to diets (increased by 1.39-fold), implying that the conversion of 2KGA to ASA in zebrafish exhibits a remarkable efficiency (with a conversion efficiency of approximately 25%). Based on the previous study, we have observed that a concentration of 5 mM-2KGA significantly increased ASA levels in *Arabidopsis thaliana* and *Brassica campestris* ssp. *chinensis* by 26.19 and 20.73%, respectively, while achieving an approximate 2KGA conversion efficiency of 8–15%. The aforementioned data demonstrate that 2KGA exhibits high conversion efficiency in both fish and plants, indicating its potential as a precursor for enhancing ASA synthesis metabolic processes. 

It is well known that most animals utilize the glucuronide pathway to synthesize ASA from D-glucose. The ASA synthesis in fish resembles that of terrestrial animals [36]. In this process, D-glucose is used as an initial precursor and is converted to L-gulono-1,4-lactone by regucalcin (also known as senescence marker protein 30 (SMP30)) after several rounds of enzymatic reaction [37]. L-gulono-1,4-lactone then participates in a final oxidation reaction catalyzed by L-gulono-1,4-lactone oxidase (GLO) and leads to the formation of ASA [38]. GLO is the key and rate-limiting enzyme in the final step of the animal’s ASA pathway [39]. In plants, on the other hand, the last step of the *myo*-inositol and L-gulose pathways is also catalyzed by GLO in the conversion of L-gulono-1,4-lactone to ASA, which is the same that in the animal pathway. As mentioned above, our team recently found direct evidence of 2KGA as a precursor to ASA conversion in plants utilizing the stable isotope tracing technique (unpublished data). Gao et al. [23] have reported that exogenous 2KGA could enhance the ASA accumulation, in which GLO played a crucial function in plants. As such, we hypothesized that the L-gulose pathway was significantly upregulated in the conversion of 2KGA to ASA in plant. However, regarding animals, it is reported that not all animals possess GLO activity for synthesizing ASA. Invertebrates, teleost fishes (including zebrafish), some passerine birds, and anthropoid primates lack the key enzyme GLO for catalyzing the final step in the ASA synthesis pathway, thereby losing their ability to synthesize ASA [40]. However, a recent discovery indicates that several cartilaginous and non-teleost bony fish species are capable of synthesizing ASA [41]. Although the current research suggests that zebrafish do not have GLO catalytic activity, it is possible that under the influence of 2KGA, zebrafish may express isozymes with GLO-like activity. This could potentially serve as a pathway for 2KGA to promote ASA accumulation in zebrafish. Therefore, it is worth proceeding with proteomics to identify and characterize the enzymes involved in the synthesis process and to delve into the mechanism of action of 2KGA on ASA synthesis in animals.

Currently, the elevation of ASA levels in aquatic animals is primarily achieved through exogenous supplementation of vitamin C in their diet [8]. Crystallized ASA is extremely unstable and most of its activity in practical diets is lost during processing and storage because of exposure to high temperatures, oxygen, and light [42]. Encapsulation or chemical modification offers a way to improve the stability of ASA [43,44]. Despite the potential benefits of encapsulated ASA, there are concerns about their biosafety for use in aquaculture environments [16]. In addition, the elevated price of these derivatives of ASA compared with the ASA also leads to a significant increase in manufacturing costs [14]. On the other hand, transgenic technology has been utilized in only a limited number of studies to demonstrate elevated levels of ASA in aquatic animals. There are existing attempts to revert GLO expression in ASA-deficient animals using transgenic methods [45,46]. However, the application of transgenic technology in aquaculture still poses certain technical difficulties, and significant attention must be paid to the ecological and genetic risks associated with transgenic aquatic animals. In our study, the increase in ASA levels resulting from 2KGA supplementation is comparable to that achieved through direct ASA supplementation in diets. 2KGA, as a precursor for industrial ASA synthesis, and utilizing it to produce a finalized ASA product, involves a series of chemical methods, including lactonization and enolization. The process of converting 2KGA to ASA is a complex operation involving the consumption of energy and a high cost [20]. As a result, the current market price of 2KGA is only one-thirds of the ASA market price, and it demonstrates chemical stability, and accessibility, and does not require chemical steps in its industrial conversion production, making it more cost-effective for market applications [47,48]. It has also been shown that natural sources of ASA have greater bioavailability than industrially synthesized ASA [49]. Therefore, in comparison to supplementation with encapsulated ASA and its derivatives, we believe that the utilization of 2KGA to stimulate the endogenous ASA biosynthesis process holds greater significance for aquatic organisms. This approach might allow quite effective management of future high-density aquaculture systems.

The ASA requirement of aquatic animals varies widely depending on the species, environment, individual growth, developmental stages, etc. Thus, the ASA synthesis capacity of the same species may vary in different growth stages [50]. However, in this study, we determined the ASA content of zebrafish only on the 21st day, i.e., a single timepoint of sampling, and focused on exploring whether 2KGA would increase the ASA level. We wondered whether 2KGA would lead to a dynamic variation in ASA content during different periods of growth. Therefore, in subsequent studies, we will investigate the dynamic changes in ASA content during various growth stages of zebrafish under 2KGA treatment and analyze the effects of 2KGA on the entire process of ASA metabolism. In this study, we selected zebrafish (*Danio rerio*) as experimental models. Today, the zebrafish has become a favored model organism for metabolic mechanisms research due to its small size, rapid life cycle, easy husbandry, and low cost, as well as because the zebrafish shares many molecular, biochemical, cellular and physiological characteristics with higher vertebrates [36]. Our research on ASA synthesis metabolism using zebrafish can therefore provide reliable data for subsequent applications in other species of animals. We will persist in investigating the impact of 2KGA on ASA biosynthesis in diverse aquatic species through future research endeavors.

Growth and development parameters are essential indicators for the quality of the aquatic product. In this study, we combined the assay of growth and development levels with a comprehensive analysis of zebrafish responses to 2KGA, providing a more integrated understanding of 2KGA application in aquaculture. In this work, the rearing conditions and the basal feed (commercial feed) formulations were selected according to the relevant literature [51,52,53,54,55], which is completely suitable for the nutritional requirements and growth of zebrafish. Meanwhile, we found that the test animals had a healthy body color, normal feeding behavior, and flexible activities after feeding with 2KGA. Our results showed that the addition of 2KGA to zebrafish feed promoted the growth of aquatic animals and increased the specific growth rate (SGR), weight gain rate (WGR), and condition factor (CF). As indicators of animal growth status, SGR, WGR, survival rate, and CF directly reflected the effect of 2KGA on the growth of aquatic animals. It is indicated that 2KGA could be used as a small molecule organic acid to benefit the performance and health of aquatic animals. Dietary organic acids are increasingly being explored as a potential strategy for enhancing growth and nutrient utilization in aquatic animals [56]. Organic acid supplementation has been reported to positively affect the growth performance of zebrafish (*Danio rerio*) [57], caspian white fish (*Rutilus kutum*) [58], and common carp [59]. This is similar to the results of our study. However, the diets supplemented with 2KGA require further refinement in terms of nutritional availability. In the current study, the rearing of zebrafish after the addition of dietary 2KGA was also found to affect whole-body proximate composition by elevating lipid and protein contents and decreasing ash contents. The enhancement of whole-body proximate composition suggested that 2KGA may affect energy metabolic pathways in aquatic animals. This indicated that 2KGA effectively improves the utilization efficiency of aquatic animal protein sources and enhances whole-body protein composition. It has been reported that organic acids showed hepatopancreatic protective properties against vibriosis in aquatic animals [60]. However, no significant differences were observed in HSI and GSI among the different treatments, which may be due to the non-stressful environment of zebrafish survival, suggesting that these indices might not be appropriate in this context.

In this study, our data showed that 2KGA significantly enhanced zebrafish feed utilization, resulting in increased feed intake (FI) and protein efficiency ratio (PER) as well as a reduced feed conversion ratio (FCR). Organic acids reportedly affect the palatability of feed for aquatic animals [61,62]. Furthermore, organic acids could decrease the pH of the feed, thus increasing pepsin activity and protein digestion [63]. Therefore, we will continue to investigate the potential of using 2KGA to formulate organic-pleasing diets for aquatic animals in the future. The correlation results demonstrate that ASA content is positively correlated with growth parameters, feeding efficiency, and whole-body protein and lipid levels, suggesting that 2KGA may influence the growth and development of aquatic animals via the regulation of ASA metabolism. Although there may be some challenges in the initial application of 2KGA, it substantially increases the ASA content in organisms, and promotes the growth and survival performance of aquatic animals with immune enhancing, anti-stress, and antioxidant properties. However, since this is a preliminary study, further investigation into the roles of 2KGA in animals is essential for its future application in aquaculture.

## 5. Conclusions

In summary, we report for the first time that 2KGA significantly increases the ASA content and facilitates the growth and health of zebrafish. Our findings imply that 2KGA may serve as a precursor for ASA synthesis, thus representing a novel alternative pathway for elevating ASA levels in aquatic animals. In addition, 2KGA may influence the growth and development of aquatic animals via the regulation of ASA metabolism. Nevertheless, its underlying mechanism requires further intensive investigation. Due to its positive effects, 2KGA supplementation of fish feedstuff should be considered a potential method for application in sustainable aquaculture.

## Figures and Tables

**Figure 1 animals-13-02502-f001:**
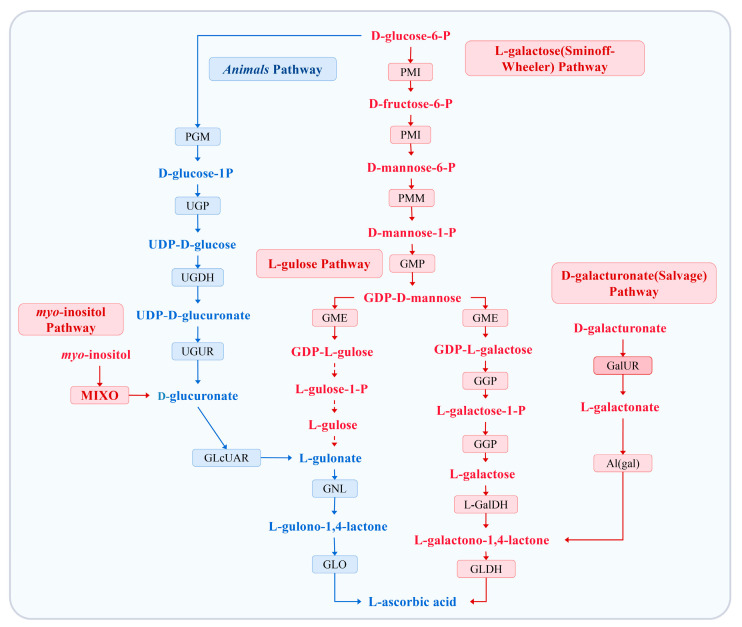
L-ascorbic acid biosynthetic pathways in plants and animals. The arrows/metabolites/enzymes depicting the four major ASA biosynthetic pathways in plants are in red, and those for ASA biosynthetic pathways in animals are in blue. Dashed arrows indicate hypothetical reactions or enzymes. PMI: phosphomannomutase; PMM: phosphomannomutase; PGM: phosphoglucomutase; GMP (VTC1): GDP-mannose pyrophosphorylase; GME: GDP-D-mannose-3′,5′-epimerase; GGP (VTC2): GDP-L-galactose phosphorylase; UGP: UDP-glucose pyrophosphorylase; UGDH: UDP-glucose dehydrogenase; UGUR: glucuronate-1-phosphate uridylyltransferase; GlcUAR: glucuronate reductase; L-GalDH: L-galactose dehydrogenase; GalUR: aldonolactonase; GLDH: L-galactono-1,4-lactone dehydrogenase; AL (gal): aldonolactonase for L-galactonate; GNL: L-gulose dehydrogenase; MIXO: *myo*-inositol monophosphatase; GLO: L-gulono-1,4-lactone oxidase.

**Figure 2 animals-13-02502-f002:**
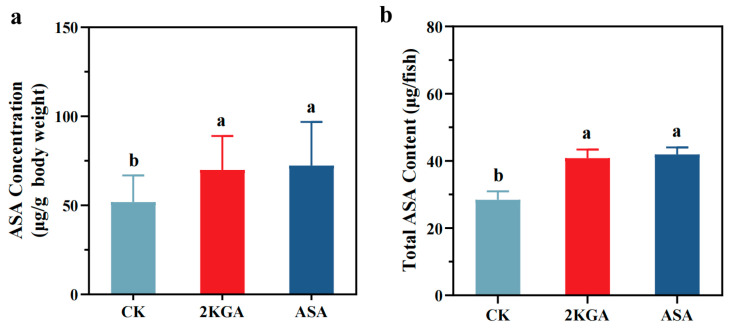
Effect of exogenous 2-keto-L-gulonic acid on the L-ascorbic acid concentration (**a**) and the total ASA content (**b**) of zebrafish. CK, control; 2KGA, 2-keto-L-gulonic acid; ASA, L-ascorbic acid. Error bars represent the mean ± SE, *n* = 3. Different letters indicate statistically significant differences (*p* < 0.05).

**Figure 3 animals-13-02502-f003:**
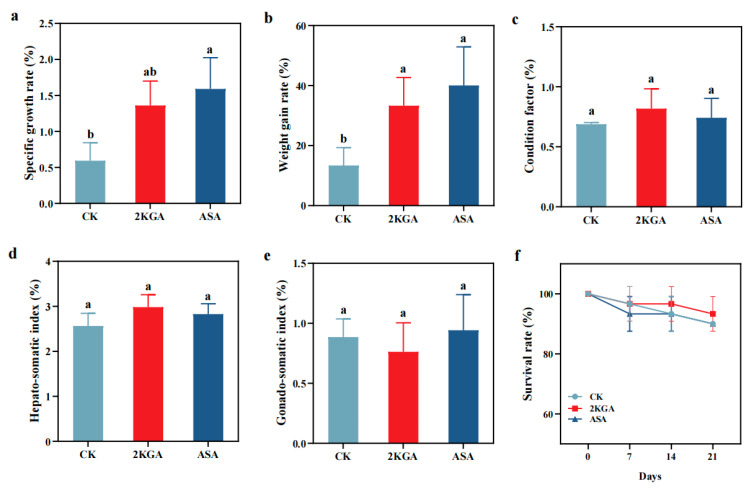
Effects of exogenous 2-keto-L-gulonic acid on growth parameters and organ-somatic indices of zebrafish. Abbreviations are as follows: (**a**) specific growth rate (SGR); (**b**) weight gain rate (WGR); (**c**) condition factor (CF); (**d**) hepato-somatic index (HSI); (**e**) gonado-somatic index (GSI); (**f**) survival rate. CK, control; 2KGA, 2-keto-L-gulonic acid; ASA, L-ascorbic acid. Different letters indicate statistically significant differences (*n* = 3, *p* < 0.05).

**Figure 4 animals-13-02502-f004:**
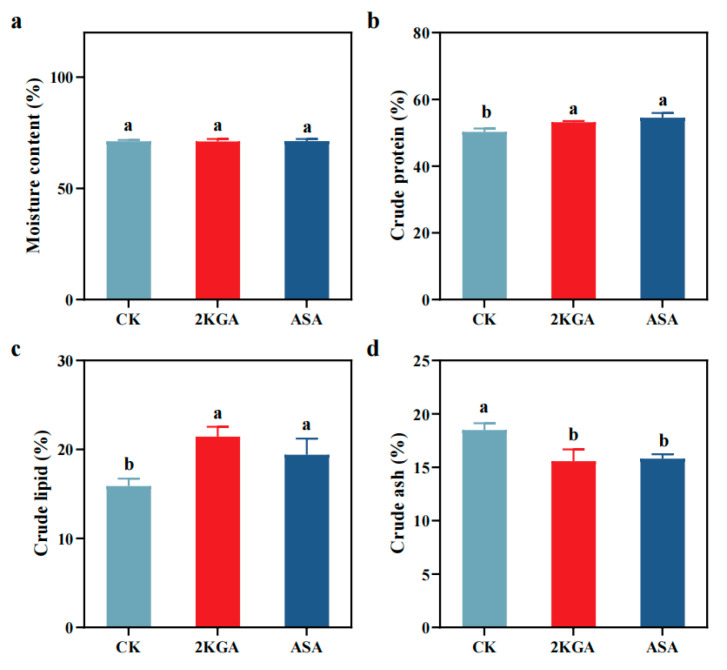
Effects of exogenous 2-keto-L-gulonic acid on whole-body proximate chemical composition of zebrafish (% dry weight). Abbreviations are as follows: (**a**), moisture content; (**b**) crude protein; (**c**) crude lipid; (**d**) crude ash. CK, control; 2KGA, 2-keto-L-gulonic acid; ASA, L-ascorbic acid. Different letters indicate statistically significant differences (*n* = 3, *p* < 0.05).

**Figure 5 animals-13-02502-f005:**
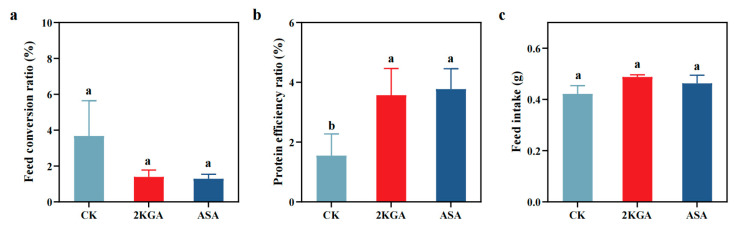
Effects of exogenous 2-keto-L-gulonic acid on feed utilization of zebrafish. Abbreviations are as follows: (**a**) feed conversion ratio (FCR); (**b**) protein efficiency ratio (PER); (**c**) feed intake (FI). CK, control; 2KGA, 2-keto-L-gulonic acid; ASA, L-ascorbic acid. Different letters indicate statistically significant differences (*n* = 3, *p* < 0.05).

**Figure 6 animals-13-02502-f006:**
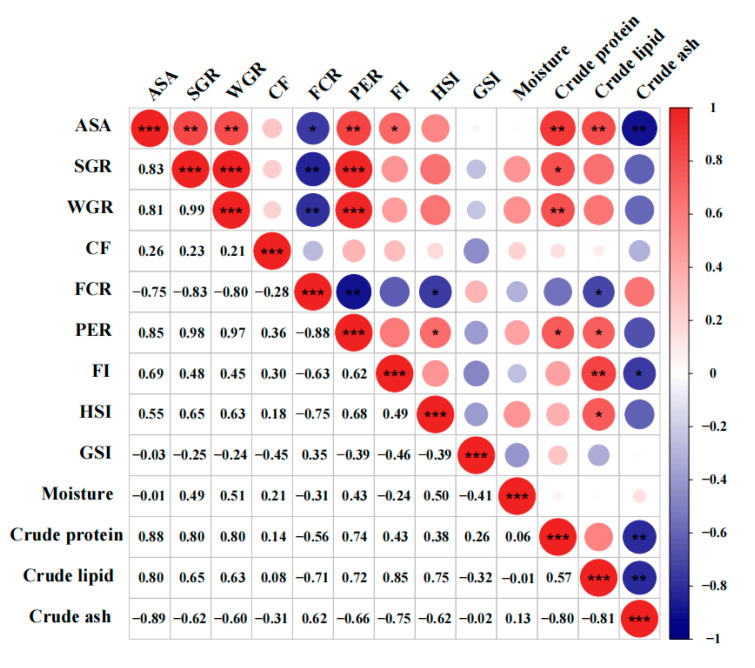
Analysis of correlation among ASA content, growth parameters and feed utilization of zebrafish (*n* = 3). Red color indicates positive correlations and blue color indicates negative correlations. Values represented the correlation coefficients. Significant differences are indicated as * *p* < 0.05, ** *p* < 0.01, *** *p* < 0.001.

**Table 1 animals-13-02502-t001:** Effect of exogenous 2-keto-L-gulonic acid on the weight (g) of zebrafish.

Parameters	Treatments		
	CK	ASA	2KGA
Initial Weight (g)	0.48 ± 0.01 ^a^	0.41 ± 0.03 ^a^	0.45 ± 0.01 ^a^
Final Weight (g)	0.54 ± 0.03 ^a^	0.56 ± 0.02 ^a^	0.60 ± 0.02 ^a^

Values are presented as mean ± SE (*n* = 3). Different letters indicate statistically significant differences (*p* < 0.05).

## Data Availability

The data that support the findings of this study are available from the corresponding author upon reasonable request.

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
