# Peer review of "Exogenous 2-keto-L-gulonic Acid Supplementation as a Novel Approach to Enhancing L-ascorbic Acid Biosynthesis in Zebrafish (Danio rerio)"

_animals, 2023, doi:10.3390/ani13152502_

Round 1
Reviewer 1 Report (Previous Reviewer 2)
All my concerns have been properly addressed. No further comments.
Author Response
Response letter:
Replies to Reviewer 1
Dear Reviewer:
We have received your positive evaluation of the manuscript entitled: “Exogenous 2-keto-L-gulonic acid supplementation as a novel approach to enhancing L-ascorbic acid biosynthesis in zebrafish (Danio rerio)”. We sincerely appreciate your acknowledgment of the value of our research efforts! Thank you once again for your constructive feedback and insightful suggestions, which will undoubtedly enhance the quality of our manuscript.
Kind regards.
Sincerely,
Hui Xu
xuhui@iae.ac.cn

Reviewer 2 Report (Previous Reviewer 4)
Dear authors,
As requested, I reviewed the manuscript (animals-2507827) “Exogenous 2-keto-L-gluconic acid supplementation as a novel approach to enhancing L-ascorbic acid biosynthesis in zebrafish (Danio rerio)” by Meijun Shi, Mingfu Gao, Hao Sun, Weichao Yang, Hongxia Zhao, Lixin Zhang and Hui Xu.
Compared to the old rejected version of the manuscript, the authors followed my suggestion and focused the analysis on just one animal model, i.e. zebrafish (Danio rerio). Despite the modification, the paper still shows some weaknesses and issues, and, for this, I suggest to reconsider the paper after major revisions. My questions/doubts are as follows:
1. My first concern regards the diets used for the supplementation in zebrafish. The formulation and the proximate composition should be provided for all the experimental diets. The authors must prove that all the diets (basal and experimental) are isoenergetic and that the concentrations of ASA or 2KGA is the only different ingredient without significantly affecting the proximate composition compared to the basal diet.
2. It is not well clear why 2KGA should be used as novel supplementation in zebrafish diet compared to ASA supplementation. Both experimental diets administered at the same concentration (800 mg/kg) show the same results in all the investigated analyses. Why should 2KGA supplementation be further explored as alternative strategy? Is it cheaper? Does it show less cost of production? It is not clear. The choice to investigate only one concentration does not help to understand the effectiveness of 2KGA. Furthermore, the authors should state why 800 mg/kg has been chosen as the only concentration investigated.
3. In the Introduction, references(s) in lines 60-61 should be added. Furthermore, a brief explanation about the use of zebrafish as animal model for the investigation of 2KGA supplementation should be added.
4. In Materials and Methods, information regarding anesthetic procedures prior to the sampling and volume of the tank where the replicates (n = 10) have been distributed should be added.
5. In Table 1 (initial and final weight), why it has been compared the mean of 5 fish total in three replicates and not the mean of single fish in three replicates? The weight of single zebrafish would be more appropriate. The same should be applied also for the other measurements.
6. In Table 1 and in Figure 1 it is not necessary to mention zebrafish. It is the only animal model used.
7. ASA concentration is expressed as µg of ASA related to g of what?
8. In general, all the results have been described according to 2KGA diet only. The results shown by zebrafish administered with ASA should be mentioned.
9. The Discussion is too long. Much of the information reported is not necessary. The authors should discuss the results in the Discussion and explain why 2KGA supplementation should be further investigated as novel approach in zebrafish although it shows the same identical results/trend of ASA supplementation
Thank you very much for your attention to my opinion.
Author Response
Response letter:
Replies to Reviewer 2
Dear Reviewer:
We have received your comments on the manuscript entitled: “Exogenous 2-keto-L-gulonic acid supplementation as a novel approach to enhancing L-ascorbic acid biosynthesis in zebrafish (Danio rerio)”. We greatly appreciate your professional review of our article. According to your valuable suggestions and comments, we have made careful revisions to the manuscript. The detailed responses are listed below and marked Red.
Point 1: My first concern regards the diets used for the supplementation in zebrafish. The formulation and the proximate composition should be provided for all the experimental diets. The authors must prove that all the diets (basal and experimental) are isoenergetic and that the concentrations of ASA or 2KGA is the only different ingredient without significantly affecting the proximate composition compared to the basal diet.
Response 1: Thank you for your constructive comments on our article. Prior to commencing the feeding experiment, we conducted a comprehensive analysis of the proximate composition of both basal and experimental diets, as presented in Table S1 below. The inclusion of ASA or 2KGA did not exert a significant impact on the proximate composition of either the basal or experimental diets, as indicated by the data presented in Table S1. The formulation and proximate composition of all experimental diets have been included in Supplementary Material Table S1, in response to your valuable suggestion.
Table S1. Formulation and proximate composition of the experiment diets (%, dry matter).
|
Experimental diets |
||
Ingredients (% dry matter) |
Control group (basal diet) |
ASA group |
2KGA group |
Fish meal |
13 |
13 |
13 |
Soybean meal |
24 |
24 |
24 |
Rapeseed meal |
22 |
22 |
22 |
Soybean oil |
6 |
6 |
6 |
Fish oil |
3.5 |
3.5 |
3.5 |
Lecithin |
2 |
2 |
2 |
Wheat middling |
20 |
20 |
20 |
Wheat gluten |
5.5 |
5.5 |
5.5 |
Mineral premixa |
2 |
2 |
2 |
Binder |
2 |
2 |
2 |
ASAb |
0 |
800 mg/kg of diet |
0 |
2KGA-NaC |
0 |
0 |
800 mg/kg of diet |
|
|
|
|
Proximate composition (% dry matter) |
Control group (basal diet) |
ASA group |
2KGA group |
Dry matter |
93.60% |
93.42% |
93.58% |
Crude protein |
38.90% |
38.84% |
38.76% |
Crude lipid |
15.02% |
15.11% |
15.08% |
Ash |
11.04% |
11.08% |
11.02% |
a Mineral premix (mg/kg of diet): MnSO4, 40 mg; MgO, 10 mg; K2SO4, 40 mg; ZnCO3, 60 mg; KI, 0.4 mg; CuSO4, 12 mg; Ferric citrate, 250 mg; Na2SeO3, 0.24 mg; Co, 0.2 mg. |
|||
b ASA (> 99.5%) were supplied by Northeast Pharmaceutical Group Co., Ltd, Shenyang, China. |
|||
C 2KGA-Na (> 99.5%) were supplied by Northeast Pharmaceutical Group Co., Ltd, Shenyang, China. |
Point 2: It is not well clear why 2KGA should be used as novel supplementation in zebrafish diet compared to ASA supplementation. Both experimental diets administered at the same concentration (800 mg/kg) show the same results in all the investigated analyses. Why should 2KGA supplementation be further explored as alternative strategy? Is it cheaper? Does it show less cost of production? It is not clear. The choice to investigate only one concentration does not help to understand the effectiveness of 2KGA. Furthermore, the authors should state why 800 mg/kg has been chosen as the only concentration investigated.
Response 2: Thank you very much for your guidance and suggestions on our research!
- We deem 2KGA to possess the subsequent benefits as an innovative supplementation:
(1)Currently, the elevation of ASA levels in aquatic animals is primarily achieved through exogenous supplementation of vitamin C in their diet [1]. However, ASA is unstable, and its oxidative degradation is accelerated in alkaline environments, and the rate of its loss is influenced by factors such as metal ions, oxygen, moisture, light, high temperature, and pH [2]. Numerous attempts have been made to improve the stability of ASA in premixes and finished feeds, mainly via encapsulation or chemical modification of ASA into its phosphate derivatives. However, the elevated price of these derivatives of ASA compared with the ASA leads to a significant increase in manufacturing costs [3]. In addition, the biosafety and environmental impact of market-encapsulated ASA in aquaculture remains to be evaluated [4]. Therefore, it is imperative to develop novel technologies that can effectively augment ASA levels in aquatic organisms.
The 2KGA is the direct precursor for industrial ASA production. 2KGA is initially biotransformed from L-sorbose by a co-culture of Ketogulonigenium vulgare and Bacillus spp. in two-step fermentation. Then, the production of a finalized ASA product through 2KGA involves a series of chemical methods, including lactonization and enolization [5]. The process of converting 2KGA to ASA is a complex operation, consumption of energy, and high cost [6]. As a result, the current market price of 2KGA is only one-third of the ASA market price. Meanwhile, 2KGA demonstrates chemical stability, accessibility, and without chemical steps in its industrial conversion production, making it more cost-effective for market applications.
In recent studies, we have demonstrated that exogenous 2KGA serves as a direct substrate for the synthesis of ASA in plants, thereby effectively promoting ASA accumulation. In addition to Arabidopsis thaliana, a significant increase in ASA levels was observed across several field crops including Brassica campestris ssp. chinensis, Spinacia oleracea, Piper nigrum, and Brassica oleracea upon treatment with 2KGA. It indicates that the 2KGA supplement may serve as a potential novel strategy to increase ASA levels in organisms. Therefore, in this study, we chose exogenous 2KGA as a supplement to investigate its potential for enhancing the growth of aquatic animals and increasing ASA accumulation.
(2)ASA is an essential nutrient for the optimal growth and development of most aquaculture species [1]. In this study, we present preliminary evidence suggesting that exogenous supplementation of 2KGA has the potential to enhance animal growth and promote the accumulation of ASA, utilizing zebrafish as a model organism. We have observed that the improvements of ASA after 2KGA supplementation are almost equivalent to those for direct ASA supplementation to diets. Many studies have shown that natural sources of ASA have greater bioavailability than that found in supplements [7]. Therefore, in comparison to supplementation with encapsulated ASA and its derivatives, we believe that the utilization of 2KGA to stimulate the endogenous ASA biosynthesis process holds greater significance for aquatic organisms.
(3) Organic acids are increasingly being investigated as dietary additives to various aquatic species as a potential means of improving their growth, nutrient utilization, and resistance to pathogenic bacteria [8]. In addition, organic acids can act as readily available energy sources or feed attractants for aquatic animals [9]. Our findings suggested that 2KGA can function as a small-molecule organic acid, effectively enhancing the performance, health, and feed utilization of aquatic animals. It also suggests that the utilization of 2KGA, a small molecule organic acid in the formulation of organic feed for aquatic animals, exhibits significant potential for practical application.
In conclusion, due to its positive effects, further investigation into the roles of 2KGA in animals is essential for its future application in aquaculture.
- In this study, we postulated that 2KGA may serve as a precursor substance in ASA accumulation. Therefore, the basal diet of the experimental group was supplemented with an equivalent amount of 2KGA-Na based on the ASA dosage in the positive control group. The ASA requirements vary widely among different fish species. The dietary supplement of ASA generally ranges from 25 to ~2,000 mg/kg and is known to improve the quality of fish [1]. After referring to reports of ASA application to zebrafish [1,10,11], we chose a moderate dosage of 800 mg/kg for this study. This study focuses on reporting the phenomenon of 2KGA for enhancing ASA in aquatic animals. Future studies will provide additional information on optimized potency at different concentration ratios for various aquatic animals.
Point 3: In the Introduction, references(s) in lines 60-61 should be added. Furthermore, a brief explanation about the use of zebrafish as animal model for the investigation of 2KGA supplementation should be added.
Response 3: Thank you very much for your helpful suggestion. We have added 4 references (Barra, et al., 2019; Yoshitomi, et al., 2004; Luisi, et al., 2021; Liu, et al., 2003) on lines 61-63, page 2. Meanwhile, we supplemented the Introduction section with a brief explanation of the use of zebrafish as an animal model for the investigation of 2KGA supplementation. The additional content is as follows: On lines 104-106, “Zebrafish (Danio rerio) is an excellent vertebrate model in various fields of aquaculture research [28]. Currently, it is being extensively utilized for fish nutritional evaluation and metabolic mechanism studies [29,30].”
Point 4: In Materials and Methods, information regarding anesthetic procedures prior to the sampling and volume of the tank where the replicates (n = 10) have been distributed should be added.
Response 4: We sincerely appreciate the valuable comments. We have added the anesthetic procedures and the volume of the tank of experimental animals to Materials and Methods 2.3 and 2.4. The additional content is as follows: On lines 148-150, page 4, “At the end of the rearing trial, all zebrafish were fasted for 24h and anesthetized with MS-222 (0.1 g/L, Sigma-Aldrich, China). Five zebrafish were randomly selected from each replicate for weighing in order to calculate growth parameter indices and feed utilization.” On lines 137-139, page 4, “Each experimental group consisted of three replicates (tanks), with ten zebrafish in each replicate. Each FRP tank is filled with 8L of water.” On lines 142-144, page 4, “All the zebrafish were maintained under the aforementioned acclimatized conditions. We conducted regular monitoring of the physicochemical parameters of the aquarium water.”
Point 5: In Table 1 (initial and final weight), why it has been compared the mean of 5 fish total in three replicates and not the mean of single fish in three replicates? The weight of single zebrafish would be more appropriate. The same should be applied also for the other measurements.
Response 5: We have taken your suggestion into full consideration and made revisions to our manuscript accordingly. Specifically, we have updated the initial and final weight data in Table 1 to reflect the weight of a single zebrafish. Apart from that, all our other measurements were evaluated based on the outcomes of a single zebrafish assay.
Point 6: In Table 1 and in Figure 1 it is not necessary to mention zebrafish. It is the only animal model used.
Response 6: Thanks for your comments. The tables and figures in the manuscript have been thoroughly examined and meticulously revised based on your recommendations.
Point 7: ASA concentration is expressed as µg of ASA related to g of what?
Response 7: Thank you for pointing this out. We have revised the related contents in our revised manuscript, including Figure 2. We added the µg of ASA related to g of body weight (µg/g body weight).
Point 8: In general, all the results have been described according to 2KGA diet only. The results shown by zebrafish administered with ASA should be mentioned.
Response 8: Thank you kindly for your guidance and suggestions on our research. We have revised the Results section to include additional experimental results from the ASA treatment group of zebrafish.
The newly added contents are as follows:
On lines 233-241, “The moisture of zebrafish did not show significant differences among treatments (Figure 4a) (p > 0.05). The 2KGA and ASA treatment significantly boosted the levels of crude protein and crude lipid in zebrafish. The crude protein contents of zebrafish treated with 2KGA and ASA increased by 5.75 and 8.52%, respectively, compared to the CK group (Figure 4b) (p < 0.05). Similarly, the crude lipid contents increased by 34.93 and 22.16%, respectively (Figure 4c) (p < 0.05). The crude ash content levels decreased in zebrafish reared under 2KGA treatments compared to the CK group (15.76% decrease) (Figure 4d) (p < 0.05). The crude ash content under ASA treatments also decreased by 14.46 % (Figure 4d) (p < 0.05).”
On lines 248-256, “The inclusion of 2KGA and ASA in the diet resulted in increased feed intake (FI) and protein efficiency ratio (PER) for zebrafish, as compared to the control group (CK) (Figure 5). The feed conversion ratio (FCR) of zebrafish decreased when 2KGA and ASA were added (61.99 and 64.84% decrease, respectively) (Figure 5a). The PER increased by 130.76 and 143.91%, respectively, in zebrafish (Figure 5b) (p < 0.05). Meanwhile, the FI of zebrafish treated with 2KGA and ASA also increased by 15.65 and 9.72%, respectively, compared to the CK group (Figure 5c). A low feed conversion ratio (FCR) is the primary indicator of efficient aquaculture because it suggests superior feed utilization efficiency for the 2KGA group in zebrafish.”
Point 9: The Discussion is too long. Much of the information reported is not necessary. The authors should discuss the results in the Discussion and explain why 2KGA supplementation should be further investigated as novel approach in zebrafish although it shows the same identical results/trend of ASA supplementation.
Response 9: I appreciate your highlighting of this deficiency. We have completely refined and amended the Discussion section in the revised manuscript. Meanwhile, we have hightlighted the advantages and necessity of 2KGA as a novel alternative strategy. The additional content is as follows: On lines 338-351, “In our study, the increase in ASA levels resulting from 2KGA supplementation is comparable to that achieved through direct ASA supplementation in diets. 2KGA, as a precursor for industrial ASA synthesis, and utilizing it to produce a finalized ASA product involves a series of chemical methods, including lactonization and enolization. The process of converting 2KGA to ASA is a complex operation, consumption of energy, and high cost [20]. As a result, the current market price of 2KGA is only one-third of the ASA market price. 2KGA demonstrates chemical stability, accessibility, and without chemical steps in its industrial conversion production, making it more cost-effective for market applications [47,48]. It has also been shown that natural sources of ASA have greater bioavailability than industrially synthesized ASA [49]. Therefore, in comparison to supplementation with encapsulated ASA and its derivatives, we believe that the utilization of 2KGA to stimulate the endogenous ASA biosynthesis process holds greater significance for aquatic organisms.”
References
- Dawood, M.; Koshio, S. Vitamin C supplementation to optimize growth, health and stress resistance in aquatic animals. Reviews in Aquaculture 2018, 10, 334-350.
- Lin, M.F.; Shiau, S.Y. Requirements of vitamin C (L-ascorbyl-2-monophosphate-Mg and L-ascorbyl-2-monophosphate-Na) and its effects on immune responses of grouper, Epinephelus malabaricus. Aquaculture Nutrition 2004, 10, 327-333.
- Yoshitomi, B. Depletion of ascorbic acid derivatives in fish feed by the production process. Fisheries science 2004, 70, 1153-1156.
- Luis, A.I.S.; Campos, E.V.R.; Oliveira, J.L.; Vallim, J.H.; Proença, P.L.F.; Castanha, R.F.; de Castro, V.; Fraceto, L.F. Ecotoxicity evaluation of polymeric nanoparticles loaded with ascorbic acid for fish nutrition in aquaculture. Journal of nanobiotechnology 2021, 19, 163.
- Yang, W.; Han, L.; Mandlaa, M.; Zhang, H.; Zhang, Z.; Xu, H. A plate method for rapid screening of Ketogulonicigenium vulgare mutants for enhanced 2-keto-L-gulonic acid production. Brazilian journal of microbiology 2017, 48, 397-402.
- Pappenberger, G.; Hohmann, H.P. Industrial production of L-Ascorbic Acid (Vitamin C) and d-Isoascorbic acid. Biotechnology of Food and Feed Additives2014; pp. 143-188.
- Vissers, M.C.; Carr, A.C.; Pullar, J.M.; Bozonet, S.M. The bioavailability of vitamin C from kiwifruit. Advances in food and nutrition research 2013, 68, 125-147.
- Ng, W.K.; Koh, C.B. The utilization and mode of action of organic acids in the feeds of cultured aquatic animals. Aquaculture 2017, 9, 342-368.
- Da Silva, B.C.; Vieira, F.d.N.; Mouriño, J.L.P.; Ferreira, G.S.; Seiffert, W.Q. Salts of organic acids selection by multiple characteristics for marine shrimp nutrition. Aquaculture 2013, 384-387, 104-110.
- Liu, D.; Gu, Y.; Pang, Q.; Han, Q.; Li, A.; Wu, W.; Zhang, X.; Shi, Q.; Zhu, L.; Yu, H.; et al. Vitamin C inhibits lipid deposition through GSK-3β/mTOR signaling in the liver of zebrafish. Fish Physiology and Biochemistry 2020, 46, 383-394.
- Chen, YJ.; Yuan, RM.; Liu, YJ.; Yang, HJ.; Liang, GY.; Tian, LX. Dietary vitamin C requirement and its effects on tissue antioxidant capacity of juvenile largemouth bass, Micropterus salmoides. Aquaculture 2015, 435, 431-436.
Kind regards.
Sincerely,
Hui Xu
xuhui@iae.ac.cn

Reviewer 3 Report (Previous Reviewer 5)
Dear Authors,
Thank you for revision of the manuscript, it is very well prepared. That is why this time I don’t have many comments. Only one suggestion and two requests for correction:
Line 159
Indicators and parameters can be presented as a bulleted list in this form or in a form of equation
· Weight gain rate (WGR, %) = 100 x [final mean weight (g) – initial mean weight (g)/initial weight
· …
Line 442
Dots are needed in case of journal abbreviations.
Line 467, 493, 533 and 542
Journal names are listed in capital letters, they should be described like the others.
Author Response
Response letter:
Replies to Reviewer 3
Dear Reviewer:
We have received your positive evaluation of the manuscript entitled: “Exogenous 2-keto-L-gulonic acid supplementation as a novel approach to enhancing L-ascorbic acid biosynthesis in zebrafish (Danio rerio)”. Special thanks for your recognition of our research work. Thank you again for your professional review to improve the quality of our manuscript. According to your valuable suggestions and comments, we have made careful revisions to the manuscript. The detailed responses are listed below and marked Red.
Point 1: Indicators and parameters can be presented as a bulleted list in this form or in a form of equation.
Weight gain rate (WGR, %) = 100 × [final mean weight (g) - initial mean weight (g)/initial weight.
Response 1: Thank you for your constructive suggestions. The indicators and parameters of this study have been harmonized and modified to the form of equations.
Point 2: Line 442 Dots are needed in case of journal abbreviations. Line 467, 493, 533 and 542, Journal names are listed in capital letters, they should be described like the others.
Response 2: We sincerely appreciate the valuable comments. We have checked each reference and revised them carefully. We have added dots after the journal abbreviations of the references in the revised manuscript. Meanwhile, we modified the journal name on Lines 478, 518, and 579 to describe it like the other journals.
Kind regards.
Sincerely,
Hui Xu
xuhui@iae.ac.cn

Round 2
Reviewer 2 Report (Previous Reviewer 4)
Dear authors,
As requested, I reviewed the manuscript “Exogenous 2-keto-L-gluconic acid supplementation as a novel approach to enhancing L-ascorbic acid biosynthesis in zebrafish (Danio rerio)” by Meijun Shi, Mingfu Gao, Hao Sun, Weichao Yang, Hongxia Zhao, Lixin Zhang and Hui Xu.
The authors have completed my requests and, for this, I suggest to accept the paper in present form.
Thank you very much for your attention to my opinion.
This manuscript is a resubmission of an earlier submission. The following is a list of the peer review reports and author responses from that submission.
Round 1
Reviewer 1 Report
Dear Authors,
My comments are attached for improvements to the manuscript.
Title: “Exogenous 2-keto-L-gulonic acid supplementation as a novel approach to enhancing L-ascorbic acid biosynthesis of aquatic animals”
The advantage of this paper is that exogenous supplementation of 2KGA can elevate ASA synthesis in aquatic animals.
Constructive Comments to the Author:
1. Please check and verify the taxonomic nomenclature changed for Pacific white shrimp (Litopenaeus vannamei) and accordingly can be modified.
2. In Introduction part, I could not see the reason and not explained the necessity of this work to take up. Please clarify it.
3. Line no. 70, sentence “As an ………..intersect. not clearly understood or looks incomplete. Please clarify it.
4. In Materials and methods Zebrafish (Danio rerio) and Pacific white shrimp (Litopenaeus vannamei) was chosen for the study why?
5. Line 103, don’t write “We” instead can be written as 2KGA-Na (800 mg kg-1 ) added.
6. Line no 108, word smashed does not looks good instead can be written as pulverized.
7. In line no. 113, what is basic diet?
8. Line no. 119, In materials & methods section, do not write the results of water quality observations only mentioned the methodology for water quality estimation.
9. In experimental design size and age of experimental animals are missing so please add it. Further what was the water volume in which animals stocked?
10. On which basis the Dose of 2KGA-Na was decided may be clarified. And how much vitamin C is required for fish and shellfish?
11. In line no. 137 Instead of “we collected” can be written as “supernatant was collected”
12. In section 2.4., what was the sample is not clear and when sample was collected after experiment or before experiment. Moreover, how many animals were sampled not mentioned.
13. In 2.5. Analysis of Growth performance and feed utilization parameters ‘Why Gonado-somatic index (GSI, %) is required in this study???
14. 2.6. Whole-body proximate chemical composition: Author has not mentioned about the sampling and when it was performed.
15. In reference section make sure that scientific names are italicised.
16. In line no. 480, Journal name is missing.
17. In line no. 495, ANIM PHYSIOL AN N…..Please check it??
Author Response
Dear Reviewer:
We have received your comments on the manuscript entitled: “Exogenous 2-keto-L-gulonic acid supplementation as a novel approach to enhancing L-ascorbic acid biosynthesis of aquatic animals”. Thank you very much for your valuable suggestions and comments! We have made revisions to the manuscript accordingly, and our response is below and marked red.
Point 1: Please check and verify the taxonomic nomenclature changed for Pacific white shrimp (Litopenaeus vannamei) and accordingly can be modified.
Response 1: Thank you for your suggestion. We checked the taxonomic nomenclature of Pacific white shrimp carefully through NCBI Taxonomy Browser. Pacific white shrimp, the Latin scientific name, Litopenaeus(Penaeus) vannamei, is a variety of prawns of the eastern Pacific Ocean. NCBI Taxonomy ID: 6689.
Point 2: In Introduction part, I could not see the reason and not explained the necessity of this work to take up. Please clarify it.
Response 2: I appreciate your highlighting this deficiency. We supplement the Introduction section with the reason and necessity for conducting this study. The additional content is as follows: On line 50-55, “However, due to the continuous development of intensive aquaculture, aquatic animals have experienced a deceleration in growth and a reduction in immunity during the culture process [2]. In addition, intensive aquaculture can lead to environmental pollution, which in turn increases the susceptibility of aquatic animals to massive disease outbreaks. This seriously hinders the sustainable development of aquaculture [3].” On line 70-73, “Even when adding 4-5 times the required amount of ASA, its effective value cannot be guaranteed. Currently, the market applications of encapsulated ASA or its derivatives are still relatively unstable, and its impact on the environment is unknown [12].” On line 86-91, “Recently, we have demonstrated through stable isotope tracer experiments that 2KGA serves as a is involved as a precursor for the synthesis of ASA in plants and also exerts a facilitative effect on plant growth and development. Whereas, whether 2KGA also impacts the accumulation levels of ASA in animals remains to be investigated.” On line 108-112, “Based on the positive role of 2KGA in the ASA synthesis of plants and the significance of ASA for the health of aquatic animals, we proposed the following hypothesis: Exogenous 2KGA is involved in the process of ASA biosynthesis in aquatic animals, thereby enhancing their growth status and resistance to environmental stresses.”
Point 3: Line no. 70, sentence “As an ………..intersect”. not clearly understood or looks incomplete. Please clarify it.
Response 3: We have made revisions based on your suggestion. The sentence has been revised to “As an essential vitamin, the biosynthetic pathway of ASA in animals intersects with that of plants and some of the synthetic genes are homologous.” on line 90-91, page 2.
Point 4: In Materials and methods Zebrafish (Danio rerio) and Pacific white shrimp (Litopenaeus vannamei) was chosen for the study why?
Response 4: Our team’s preliminary experiments have demonstrated that 2KGA significantly enhances the accumulation of ASA in various plant species, including Arabidopsis thaliana, Brassica campestris L. ssp. chinensis, Solanum lycopersicumm, and Capsicum annuum L., etc. Therefore, we sought to investigate whether 2KGA has a universal effect on promoting ASA accumulation in animals, particularly aquatic animals. To the end, we selected two different species of aquatic experimental animals: zebrafish and Pacific white shrimp. Zebrafish are one of the significant model vertebrates in biological research, due to their distinct genetic background and high molecular, cellular, and tissue-level similarity to mammals. Pacific white shrimp is a principal species in worldwide aquaculture with high commercial value. It has also emerged as a model in the field of crustacean biology, especially because it is one of the pioneering decapod crustaceans to have undergone genome sequencing. The model aquatic animals of two distinct species were carefully chosen to provide a genetic and physiological basis for our thorough study of the mechanism of ASA synthesis. Undoubtedly, we will persist in investigating the impact of 2KGA on ASA biosynthesis in diverse aquatic species through future research endeavors.
Point 5: Line 103, don’t write “We” instead can be written as 2KGA-Na (800 mg kg-1 ) added.
Response 5: Thank you for your suggestion. The sentence has been revised to “2KGA-Na (800 mg/kg) was supplemented to the basal diet for the treatment group, while ASA (800 mg/kg) was added to the basal diet for the positive control group (2KGA-Na (> 99.5%) and ASA (> 99.5%) were supplied by Northeast Pharmaceutical Group Co., Ltd., Shenyang, China).” (line 127-130, page 4)
Point 6: Line no 108, word smashed does not looks good instead can be written as pulverized.
Response 6: Thank you for the helpful suggestion. This sentence has been revised to “The ingredients were pulverized to a particle size that could pass through an 80 μm sieve, thoroughly mixed, and then extruded into 1.0-1.5 mm pellets using a pelletizer. Afterward, they were dried at 60°C for 12h before being stored at -20°C until application.” (line 131-134, page 4)
Point 7: In line no. 113, what is basic diet ?
Response 7: The basal diet is the original feed without any addition of 2KGA and ASA.
Point 8: Line no. 119, In materials & methods section, do not write the results of water quality observations only mentioned the methodology for water quality estimation.
Response 8: We have fully considered your suggestion. The data of water quality observations have been removed from the revised manuscript.
Point 9: In experimental design size and age of experimental animals are missing so please add it. Further what was the water volume in which animals stocked?
Response 9: Thank you for your constructive suggestions. We have added the size and age of experimental animals to Materials and Methods 2.1 (line 121-123, page 3). Their initial weights are also shown in Table 1. The water volume in which animals are stocked has been included in Materials and Methods 2.2 (line 141, page 4).
Point 10: On which basis the Dose of 2KGA-Na was decided may be clarified. And how much vitamin C is required for fish and shellfish.
Response 10: In this research, we hypothesized that exogenous 2KGA may be involved in the ASA biosynthesis as a precursor and thus the diet of the 2KGA group was supplemented with an equal level of 2KGA-Na according to the ASA dosage in the positive control group. The recommended ASA supplementation in the feed of aquatic animals has been reported to vary widely, ranging from 10 to 10,000 mg/kg. However, based on the literatures, the most common values were 250-1000 mg/kg for Penaeus chinensis, 1000 mg/kg for kuruma shrimp (Marsupenaeus japonicas), and 600 mg/kg for Ctenopharyngodon idellus. The requirements for ASA vary depending on species, size, diet and farming conditions. After referring to the reports, we have selected a moderate feed supplementation of 800 mg/kg for this study. Following your comment, we have added information about the ASA requirements for fish and shrimp as well as two references in the Introduction on line 62-67, page 2.
Point 11: In line no. 137 Instead of “we collected” can be written as “supernatant was collected”.
Response 11: Thank you for pointing this out. We have revised this sentence “After centrifugation at 6000 r/min for 10 min, the supernatant was collected for the determination of ASA content by means of high-performance liquid chromatography (HPLC).” (line 163-166, page 4)
Point 12: In section 2.4., what was the sample is not clear and when sample was collected after experiment or before experiment. Moreover, how many animals were sampled not mentioned.
Response 12: We have revised this section with the hope that the details of the samples are visible now. The additional content is as follows: On line 156-158, “At the end of rearing trial, the diet was suspended for 24h, and 5 zebrafish or 5 shrimp were randomly selected from each replicate for weighing to calculate growth parameter indices and feed utilization. Subsequently, ASA measurements were taken.”
Point 13: In 2.5. Analysis of Growth performance and feed utilization parameters ‘Why Gonado-somatic index (GSI, %) is required in this study???
Response 13: The gonad-somatic index reflects the degree of gonad development and reproductive capacity of aquatic animals, while ASA also plays a significant role in aquatic animal reproduction. It has been reported that enhancing ASA levels can lead to weight gain and improved feed conversion rate of aquatic animals, and optimal maintenance of reproductive status in aquatic animals. ASA deficiency reduced both sperm concentration and motility, and thus fertility, of aquatic animals. Therefore, we examined the indicators related to ASA elevation, including GSI.
Point 14: 2.6. Whole-body proximate chemical composition: Author has not mentioned about the sampling and when it was performed.
Response 14: Thanks for your comments. We have appended the sampling details and timing in Materials and Method 2.6. The additional content is as follows: On line 185-186, “Three weeks later, three zebrafish or three shrimp were randomly selected from each replicate for measurement of their whole-body proximate chemical composition.”
Point 15: In reference section make sure that scientific names are italicised.
Response 15: We have fully considered your suggestion. We have modified the scientific names of the references to be italicized in our revised manuscript.
Point 16: In line no. 480, Journal name is missing.
Response 16: We have refilled the name of the journal on line 525.
Point 17: In line no. 495,ANIM PHYSIOL AN N.....Please check it?
Response 17: We apologize for the incorrect spelling. We have recorrected this mistake in our revised manuscript.
With best regards,
Hui Xu
xuhui@iae.ac.cn

Reviewer 2 Report
This manuscript titled “Exogenous 2-keto-L-gulonic acid supplementation as a novel approach to enhancing L-ascorbic acid biosynthesis of aquatic animals” described a very interesting study. This manuscript is very well written. However, there is some issues which need to be clarified. If this study included some gene/protein expression results, it would provide more insights.
Specific comments:
1. Line 56-57, ASA is unstable, but it is added to fish feed as ASA-phosphate.
2. Line 60-74, How is 2KGA converted into L-gulono-1,4-lactone, the substrate of the final step of ASA biosynthesis? This is a key information, and should be introduced.
3. Line 105, as mentioned above, when Vc was added into fish feed, usually it is added as Vc-phosphate. Also, please unify the name between ASA and vitamin C.
4. Line 110, does “(dry 110 weight %)” mean drying the ingredients before feed preparation?
5. Also, line 114, is 38.9% expressed as % dry matter?
6. This feed formulation quality is low. The fish and shrimp may not be at a normal growth status. The fishmeal level is low and no other additional vitamins are added. Animals at a sub-normal growth status cannot guarantee a convincible result.
7. Please add the survival data.
8. The equation “Feed efficiency (FE, g/g) = Feed consumed (g)/weight gain (g)” is wrong. Feed conversion ratio = Feed consumed (g)/weight gain (g).
9. Feed intake actually should be a ratio of Feed intake (FI, g) relative to fish weight. Please reference to some fish nutrition articles.
10. Please add unit for figure 3 and possibly other figures.
11. Figure 5, is the proximate composition expressed as % dry weight or % wet weight. They should be expressed as % wet weight.
12. Figure 6, if the Feed efficiency (FE, g/g) is calculated by Feed consumed (g)/weight gain (g), a lower FER value indicates positive effects. It means we need less feeds to produce a certain amount of fish.
13. Line 277, replace “L-Gulono-1,4-lactone” with “L-gulono-1,4-lactone”
14. Line 290-293, why not analyze the gene and/or protein expression of GLO? This is quite simple. This study should include some bio-pathway studies.
15. Line 377 and thereafter, what about the survival? The effects on fish survival should be introduced.
16. Line 373, gain the feed efficiency should be feed conversion ratio (FCR)
Author Response
Replies to Reviewer 2
Dear Reviewer:
We have received your comments on the manuscript entitled: “Exogenous 2-keto-L-gulonic acid supplementation as a novel approach to enhancing L-ascorbic acid biosynthesis of aquatic animals”. Thank you very much for your valuable suggestions and comments! We have made revisions to the manuscript accordingly, and our response is below and marked red.
Point 1: Line 56-57, ASA is unstable, but it is added to fish feed as ASA-phosphate.
Response 1: As your mentioned, ASA is prone to degradation during processing and storage due to its instability and oxidation. Currently, encapsulated ASA or ASA phosphate derivatives are frequently used to improve the stability of ASA. For this study, however, we used the crystalline ASA for the following main reasons:
- In this study, we focused on exploring the investigation of 2KGA to promote ASA synthesis. Although the chemical properties of ASA derivatives are better stabilized, the influence of their additional radicals on aquatic animals is not clear. To eliminate this factor, ASA was selected for feeding experiments to exclude the effect of phosphate groups on aquatic animals in this study.
- The biological potency of ASA dosage forms in various aquatic animals is not entirely consistent. Our research revealed that the exogenous addition of 2KGA was capable of increasing the accumulation of the endogenous ASA in aquatic animals and reached the increased level of the ASA group. However, ASA derivatives differ from crystalline ASA in that they are chemically modified on the carbocyclic chain. Therefore, we chose the crystalline ASA for a more precise and systematic comparison with the ASA levels of the 2KGA group.
- As ASA in various dosage forms is synthesized by chemical interaction, the effects of its encapsulated lipids or derived moieties on the aquatic animal environment are currently unknown. To exclude the influence of environmental factors on aquatic animals, the crystalline ASA was chosen for feeding experiments in our research.
Point 2: Line 60-74, How is 2KGA converted into L-gulono-1,4-lactone, the substrate of the final step of ASA biosynthesis? This is a key information, and should be introduced.
Response 2: Thank you kindly for your guidance and suggestions on our research. We apologize for the unclear formulation concerning the conversion of L-gulono-1,4-lactone by 2KGA. 2KGA is a precursor for the industrial synthesis of ASA, but its in vivo conversion has not been reported. Our team recently found direct evidence of 2KGA as a precursor to ASA conversion in plants utilizing the stable isotope tracing technique. Thus, we hypothesized the metabolic flow: 2KGA was first converted to L-gulonic acid, followed by further conversion to L-gulono-1,4-lactone, and eventually resulting in the production of ASA. However, the key enzyme for the conversion of 2KGA to ASA in this pathway remains uncharacterized. Therefore, we will continue to conduct further research into the metabolic mechanism of 2KGA conversion to ASA. Following your comments, we have supplemented the metabolic pathway information in the discussion section on line 327-329 “As mentioned above, our team recently found direct evidence of 2KGA as a precursor to ASA conversion in plants utilizing the stable isotope tracing technique.” On line 331-334 “Thus, we hypothesized the metabolic flow: 2KGA was first converted to L-gulonic acid, followed by further conversion to L-gulono-1,4-lactone, eventually resulting in the production of ASA. However, the key enzyme for the conversion of 2KGA to ASA in this pathway remains uncharacterized.”
Point 3: Line 105, as mentioned above, when Vc was added into fish feed, usually it is added as Vc-phosphate. Also, please unify the name between ASA and vitamin C.
Response 3: Thank you for your constructive suggestions. We have already explained the dosage form of ASA in response 1. Meanwhile, we have standardized the ASA phrasing in the revised manuscript.
Point 4: Line 110, does “(dry 110 weight %)” mean drying the ingredients before feed preparation?
Response 4: “(dry 100 weight %)” refers to the dry matter weight of the organic matter obtained by drying the feed ingredients and removing the adsorbed water. Prior to feeding experiments, the diets were prepared using fresh feeds based on dry matter weight calculation.
Point 5: Also, line 114, is 38.9% expressed as % dry matter?
Response 5: The crude protein content was 38.9% of dry matter weight (line 139, page 4).
Point 6: This feed formulation quality is low. The fish and shrimp may not be at a normal growth status. The fishmeal level is low and no other additional vitamins are added. Animals at a sub-normal growth status cannot guarantee a convincible result.
Response 6: Thank you very much for your helpful suggestion. In our research, the basal diet was composed of 13% fish meal (Animal protein source), 24% soybean meal, and 22% rapeseed meal (Plant protein source) to achieve a total crude protein content of 42%. According to the current nutritional requirements of aquatic diets, the recommended range for total protein is between 23% to 55%. Therefore, the protein content of our basal diet meets the necessary criteria for optimal growth and development of fish and shrimp. Although fishmeal has traditionally been the primary source of protein in practical fish diets, plant proteins are gradually beginning to replace fishmeal due to environmental constraints. Meng et al. [1] reported that the use of only 10% fishmeal and licorice additive exhibited better effects on the growth performance and muscle quality of carp. In the study, we observed that the fish and shrimp maintained a healthy state throughout the experimental period. In addition, we did not supplement with other vitamins, in order to maintain the focus of this research on detecting variations in ASA content. As a result, no additional vitamins were included. Moreover, the fish meal and other substances in the diet contain part of vitamin A or vitamin D, which can also provide vitamins to the animals.
Point 7: Please add the survival data.
Response 7: Thank you for your constructive comments, which are very valuable to us. We have supplemented the survival data in Figure 3 and Figure 4.
Point 8: The equation “Feed efficiency (FE, g/g) = Feed consumed (g)/weight gain (g)” is wrong. Feed conversion ratio = Feed consumed (g)/weight gain (g).
Response 8: Thank you for pointing out our error. We have corrected this in Materials and Method 2.5.
Point 9: Feed intake actually should be a ratio of Feed intake (FI, g) relative to fish weight. Please reference to some fish nutrition articles.
Response 9: Thank you for your comments. Feed intake (FI, g) is the amount of feed given or supplied during the experimental period after excluding the uneaten feed by collecting and drying the uneaten feed after half hour of each meal. We were referencing the method of detection of Feed intake in the following literature:
[1] Mansour, A.T.; Fayed, W.M.; Alsaqufi, A.S.; Aly, H.A.; Alkhamis, Y.A.; Sallam, G.R. Ameliorative effects of zeolite and yucca extract on water quality, growth performance, feed utilization, and hematobiochemical parameters of European seabass reared at high stocking densities. Aquaculture Reports 2022, 26, 101321.
[2] Díaz-Rúa, A.; Chivite, M.; Comesaña, S.; Velasco, C.; Valente, L.M.P.; Soengas, J.L.; Conde-Sieira, M. The endocannabinoid system is affected by a high-fat-diet in rainbow trout. Hormones and behavior 2020, 125, 104825.
[3] Ytrestøyl, T.; Struksnaes, G.; Koppe, W.; Bjerkeng, B. Effects of temperature and feed intake on astaxanthin digestibility and metabolism in Atlantic salmon, Salmo salar. Comparative biochemistry and physiology. Part B, Biochemistry & molecular biology 2005, 142, 445-455.
Point 10: Please add unit for figure 3 and possibly other figures.
Response 10: Thanks for your suggestion, we have added the units in Figure 3-7.
Point 11: Figure 5, is the proximate composition expressed as % dry weight or % wet weight. They should be expressed as % wet weight.
Response 11: The whole-body proximate composition was detected based on the dry weight. We have added the annotated information in Figure 5.
Point 12: Figure 6, if the Feed efficiency (FE, g/g) is calculated by Feed consumed (g)/weight gain (g), a lower FER value indicates positive effects. It means we need less feeds to produce a certain amount of fish.
Response 12: We sincerely appreciate the valuable comments. We have changed “Feed efficiency (FE)” to “Feed conversion ratio (FCR)” and supplemented the results of FCR in Result 3.3. “A low FCR is the primary indicator of efficient aquaculture. It means greater efficiency in feed utilization for the 2KGA group.” (line 287-289, page 9)
Point 13: Line 277, replace “L-Gulono-1,4-lactone” with “L-gulono-1,4-lactone”.
Response 13: “L-Gulono-1,4-lactone” has been revised to “L-gulono-1,4-lactone” (line 324).
Point 14: Line 290-293, why not analyze the gene and/or protein expression of GLO? This is quite simple. This study should include some bio-pathway studies.
Response 14: We are very appreciative of your suggestion. This study presents preliminary findings on the potential of 2KGA to enhance ASA accumulation in aquatic animals. Your valuable suggestions are important to our further study on 2KGA. ASA synthesis in animals and plants share similar routes of metabolism, involving the GLO-catalyzed synthesis of ASA from L-gulono-1,4-lactone. Previous research by Gao et al. [5] has demonstrated that exogenous application of 2KGA can upregulate GLO expression in plants, leading to increased ASA production. Although the current research suggested that zebrafish and shrimp do not have GLO catalytic activity, it is possible that under the effect of 2KGA, zebrafish and shrimp may express isozymes with GLO-like activity. This could potentially serve as a pathway for 2KGA to promote ASA accumulation in zebrafish and shrimp. Therefore, we will proceed to use proteomics to identify and characterize the synthesis enzymes of the process and delve into the mechanism of action of 2KGA on ASA synthesis in animals.
Point 15: Line 377 and thereafter, what about the survival? The effects on fish survival should be introduced.
Response 15: Thank you for your constructive comments. We have supplemented the survival data in result 3.3. On the line 234-236, “Compared to the CK group (90%), the survival rate of the 2KGA and ASA groups exhibited higher survival rates at 93% and 90%, respectively (Figure 3f) (p > 0.05)”. On the line 254-255, “The survival rates of the 2KGA and ASA groups were both 86.87%, compared to the CK group (83.33%) (Figure 4f) (p > 0.05)”.
Point 16: Line 373, gain the feed efficiency should be feed conversion ratio (FCR).
Response 16: Thank you for your constructive suggestion, “Feed efficiency” has been revised to “Feed conversion ratio” in the revised manuscript.
With best regards,
Hui Xu
xuhui@iae.ac.cn
References
- Meng, X.L.; You, F.; Cao, H.; Cai, H.M.; Li, Y.; Yang, G.K.; Zhang, Y.M.; Chang, X.L.; Zhang, X.D.; Tian, X. Effects of dietary licorice (Glycyrrhiza uralensis) supplementation on growth performance, muscle quality, and immunity in the common carp (Cyprinus carpio haematopterus). Aquaculture Reports 2022, 27, 101331.
- Mansour, A.T.; Fayed, W.M.; Alsaqufi, A.S.; Aly, H.A.; Alkhamis, Y.A.; Sallam, G.R. Ameliorative effects of zeolite and yucca extract on water quality, growth performance, feed utilization, and hematobiochemical parameters of European seabass reared at high stocking densities. Aquaculture Reports 2022, 26, 101321.
- Díaz-Rúa, A.; Chivite, M.; Comesaña, S.; Velasco, C.; Valente, L.M.P.; Soengas, J.L.; Conde-Sieira, M. The endocannabinoid system is affected by a high-fat-diet in rainbow trout. Hormones and behavior 2020, 125, 104825.
- Ytrestøyl, T.; Struksnaes, G.; Koppe, W.; Bjerkeng, B. Effects of temperature and feed intake on astaxanthin digestibility and metabolism in Atlantic salmon, Salmo salar. Comparative biochemistry and physiology. Part B, Biochemistry & molecular biology 2005, 142, 445-455.
- Gao, M.; Sun, H.; Shi, M.; Wu, Q.; Ji, D.; Wang, B.; Zhang, L.; Liu, Y.; Han, L.; Ruan, X.; et al. 2-Keto-L-Gulonic Acid Improved the Salt Stress Resistance of Non-heading Chinese Cabbage by Increasing L-Ascorbic Acid Accumulation. Front Plant Sci 2021, 12, 697184.

Reviewer 3 Report
1. Line 36-37 abstract. Say if there were any statistical differences.
2. During the acclimatization period, was the control feed used for all the experimental variants?
3. Line 109. Applying the granulation/extrusion process at a temperature of 60 degrees, are none of the researched compounds destroyed?
4. Line 123. it should be detailed if the experimental variants were starved a few days before the start of the experiment.
6. Line 195-196 Figure 2 shows 4 figures a and b for zebrafish and c and d for vaniamei.
When were the analyzes in figure a and c done?
The study is up-to-date, but I think that the experimental period is too short and the analyzes that show that the inclusion of 2-keto-L-gulonic 3 acid in the feeding of aquatic animals must be diversified. I think that some blood tests, oxidative stress and intestinal microbiota should have been done. It was not explained how 2-keto--L-gulonic could affect vitamin c nor how vitamin c affects growth parameters.
I think the experiment should be repeated with a larger number of individuals
Author Response
Replies to Reviewer 3
Dear Reviewer:
We have received your comments on the manuscript entitled: “Exogenous 2-keto-L-gulonic acid supplementation as a novel approach to enhancing L-ascorbic acid biosynthesis of aquatic animals”. Thank you very much for your valuable suggestions and comments! We have made revisions to the manuscript accordingly, and our response is below and marked red.
Point 1: Line 36-37 abstract. Say if there were any statistical differences.
Response 1: We sincerely appreciate the valuable comments. We have added statistical analysis in the abstract on line 32-38, “Our results show that 2KGA significantly increased the ASA content in zebrafish and shrimp compared with the CK group (increased by 34.82 and 37.19%, respectively, p < 0.05), reaching a consistent level with the ASA group (increased by 39.61 and 35.83%, respectively, p < 0.05). In addition, 2KGA significantly improved zebrafish- and shrimp-relevant growth parameters (Specific growth rate increased by 129.04% and 79.55%, respectively, p < 0.05) and feed utilization (Feed intake increased by 15.65% and 16.65%, respectively, p < 0.05).”
Point 2: During the acclimatization period, was the control feed used for all the experimental variants?
Response 2: All of the experimental animals were fed a basal diet for the entire domestication period.
Point 3: Line 109. Applying the granulation/extrusion process at a temperature of 60 degrees, are none of the researched compounds destroyed?
Response 3: We examined the 2KGA and ASA content of the experimental diets by HPLC prior to the feeding experiment. The 2KGA content was 798.69 mg/kg in the 2KGA group and the ASA content was 795.98 mg/kg in the ASA group. The drying temperature of 60°C did not affect the chemical properties of 2KGA and ASA.
Point 4: Line 123. it should be detailed if the experimental variants were starved a few days before the start of the experiment.
Response 4: Thank you for your constructive suggestions. Before the feeding experiment, no starvation treatment was carried out. Prior to sampling, a 24-hour period of starvation was imposed on the experimental fish and shrimp. We have supplemented in Materials and Methods 2.4. “At the end of the rearing trial, a 24-hour diet suspension was implemented and five zebrafish or five shrimp were randomly selected from each replicate for weighing to calculate growth parameter indices and feed utilization.” (line 159-162)
Point 5: Line 195-196 Figure 2 shows 4 figures a and b for zebrafish and c and d for vaniamei. When were the analyzes in figure a and c done?
Response 5: Thank you for your suggestions. We have modified Figure 2. The ASA content of zebrafish was determined by HPLC analysis following growth index measurements, as shown in Figures 2a and c.
Point 6: The study is up-to-date, but I think that the experimental period is too short and the analyzes that show that the inclusion of 2-keto-L-gulonic 3 acid in the feeding of aquatic animals must be diversified. I think that some blood tests, oxidative stress and intestinal microbiota should have been done. It was not explained how 2-keto--L-gulonic could affect vitamin c nor how vitamin c affects growth parameters.
I think the experiment should be repeated with a larger number of individuals.
Response 6: We appreciate your positive comments on our work. Our response is provided below.
- The experimental period of this study was designed to last 21 days. Based on the preliminary data, we have observed that 2KGA significantly enhances ASA accumulation in plants, with the highest level being reached on day 21. Therefore, a feeding experiment lasting 21 days was selected for this study. It has also been reported in the literature that short-term ASA supplementation can improve the nutrition level of aquatic animals.For example, Liu et al. [1] reported that ASA exerted the lipid-lowering effect through GSK-3β/β-catenin signaling in zebrafish for 14 days. López et al. [2] reported that higher blood protein, total blood cells, granular cells, and ProPO activity were recorded in shrimp fed with ASA for 40 days.
- Following your suggestion, we will subsequently conduct further experiments to investigate the effects of 2KGA on blood tests, oxidative stress, and intestinal microbiota in aquatic animals.
- It remains unclear how 2KGA participates in the synthesis of ASA in animals. Thus far, only 2KGA has been identified as a direct precursor involved in ASA biosynthesis in plants. We speculate that stable isotope techniques can be employed to trace the metabolic flow from 2KGA to L-gulonic acid, followed by an esterification reaction leading to L-gulono-1,4-lactone and ultimately culminating in ASA synthesis catalyzed by GLO. However, the key enzymes of this pathway in animals have not been characterized yet, and this is one of the research areas that we will focus on subsequently. We demonstrated a positive correlation between ASA and growth parameters utilizing correlation analysis, and we speculate that it might be that 2KGA promotes the synthesis of ASA, which in turn increases its growth parameters. We explained this phenomenon adequately in the discussion.
- In our study, which is currently in the basic research stage, 30 experimental animals per group could fulfill the basic experimental requirements. And we performed a rigorous statistical analysis of the data for each group.
With best regards,
Hui Xu
xuhui@iae.ac.cn
References
- Liu, D.; Yu, H.; Pang, Q.; Zhang, X. Investigation of the Lipid-Lowering Effect of Vitamin C Through GSK-3β/β-Catenin Signaling in Zebrafish. Frontiers in physiology 2018, 9, 1023.
- López, N.; Cuzon, G.; Gaxiola, G.; Taboada, G.; Valenzuela, M.; Pascual, C.; Sanchez, A.; Rosas, C. Physiological, nutritional, and immunological role of dietary glucan and ascorbic acid 2-monophosphate in Litopenaeus vannamei juveniles. Aquaculture 2003, 224, 223-243.

Reviewer 4 Report
Dear authors,
As requested, I reviewed the manuscript “Exogenous 2-keto-L-gluconic acid supplementation as a novel approach to enhancing L-ascorbic acid biosynthesis of aquatic animals” by Meijun Shi, Mingfu Gao, Hao Sun, Weichao Yang, Lixin Zhang and Hui Xu.
The manuscript deals with the field of the journal. The results look very interesting and promising. However, the work shows some flaws and weaknesses, especially in the experimental design, and, for this, I suggest to accept the paper with major revisions, which are as follows:
1. My primary doubt concerns the comparison of the results between zebrafish (a teleost) and Pacific white shrimps (Arthropoda). Based on the paper, the authors state that the two animals have been raised under the same rearing conditions and both have been administered with the same basal diet (alone or in combination with ASA and 2KGA), regardless of different habits, rearing conditions, wellness and nutritional requirements that the two animals show. Therefore, the authors should clarify why zebrafish and Pacific white shrimps have been chosen as animal models of the work and, above all, as the two comparison models. The authors should justify why both animals have been administered with the same basal diet and the authors should state whether the basal diet satisfy nutritional requirements of both zebrafish and Pacific white shrimps (the authors should also mention whether the basal diet is a laboratory or a commercial diet); I might suggest that the authors split the work and focus on just one animal model;
2. In general, the authors should improve the text with more references. For example, reference(s) should be added in the Introduction and for the equations in the paragraph 2.5 Analysis of growth performance and feed utilization parameters;
3. Regarding Material and Methods, the authors should add more detailed information about rearing conditions like developmental stage (it has never been mentioned if the animals are adult, juvenile, etc.), photoperiod, strain, frequency of feeding, volume of the tanks where the tested animal were distributed and the anesthesia procedures used for the sampling (as mentioned above, the authors should explain why both animals have been raised under the same rearing conditions). Furthermore, the authors should clarify why they followed the procedure of extraction and determination of ASA used in plants in a previous work. The line 143-144 “three weeks later, three replicate independent samples were collected from each experimental group, of which each sample was pooled from five zebrafish or shrimp” is not clear. Finally, in 2.6. Data analysis the authors should mention the post-hoc comparison test used for the ANOVA;
4. Based on my experience and literature, I have doubts that adult zebrafish can weigh 2 g and even more;
5. In all the legends to the figures, the statistical analysis used and the number of individuals should be always mentioned. Furthermore, in some figures, results expressed with different units of measurement have been joined in the same graph and this is not correct. It is not necessary to draw on the graph the absence of statistical difference. The description of the results of feed intake (FI) does not match the graph. Finally, the authors should specify that the correlation analysis has been performed for the 2KGA analysis only.
Thank you very much for your attention to my opinion.
Author Response
Replies to Reviewer 4
Dear Reviewer:
We have received your comments on the manuscript entitled: “Exogenous 2-keto-L-gulonic acid supplementation as a novel approach to enhancing L-ascorbic acid biosynthesis of aquatic animals”. Thank you very much for your valuable suggestions and comments! We have made revisions to the manuscript accordingly, and our response is below and marked red.
Point 1: My primary doubt concerns the comparison of the results between zebrafish (a teleost) and Pacific white shrimps (Arthropoda). Based on the paper, the authors state that the two animals have been raised under the same rearing conditions and both have been administered with the same basal diet (alone or in combination with ASA and 2KGA), regardless of different habits, rearing conditions, wellness and nutritional requirements that the two animals show. Therefore, the authors should clarify why zebrafish and Pacific white shrimps have been chosen as animal models of the work and, above all, as the two comparison models. The authors should justify why both animals have been administered with the same basal diet and the authors should state whether the basal diet satisfy nutritional requirements of both zebrafish and Pacific white shrimps (the authors should also mention whether the basal diet is a laboratory or a commercial diet); I might suggest that the authors split the work and focus on just one animal model;
Response 1:
We appreciate your positive comments on our work. Our response is below.
- In our team's previous study, we discovered that 2KGA universally enhances vitamin C accumulation in plants, including Arabidopsis thaliana, Solanum Lycopersicum, Piper nigrum Linn crops, especially the ASA of Brassica campestris increased by 188.94%. Therefore, it is necessary to further investigate whether 2KGA also promotes ASA accumulation in animals. In this study, zebrafish and Pacific white shrimp were used as model organisms. Zebrafish is one of the prominent model vertebrates in biological research due to its well-defined genetic background and high genomic similarity with mammals. Meanwhile, pacific white shrimp, as the main species of shrimp farming worldwide, has a great commercial value. It has also become a model animal for arthropoda biology research, and as a decapod arthropoda, it also has a clear genetic background. The choice of these two distinct species of model aquatic animals will not only verify the prevalence of 2KGA enhancement in aquatic animals but also provide a genetic and physiological basis for an extensive investigation into the mechanism of ASA synthesis. In future studies, we will further investigate the impact of 2KGA on ASA synthesis in other species of aquatic animals.
- In our team's previous study, we discovered that 2KGA universally enhances vitamin C accumulation in plants, including Arabidopsis thaliana, Solanum Lycopersicum, Piper nigrum Linn crops, especially the ASA of Brassica campestris increased by 188.94%. Therefore, it is necessary to further investigate whether 2KGA also promotes ASA accumulation in animals. In this study, zebrafish and Pacific white shrimp were used as model organisms. Zebrafish is one of the prominent model vertebrates in biological research due to its well-defined genetic background and high genomic similarity with mammals. Meanwhile, pacific white shrimp, as the main species of shrimp farming worldwide, has a great commercial value. It has also become a model animal for arthropoda biology research, and as a decapod arthropoda, it also has a clear genetic background. The choice of these two distinct species of model aquatic animals will not only verify the prevalence of 2KGA enhancement in aquatic animals but also provide a genetic and physiological basis for an extensive investigation into the mechanism of ASA synthesis. In future studies, we will further investigate the impact of 2KGA on ASA synthesis in other species of aquatic animals.
Point 2: In general, the authors should improve the text with more references. For example, reference(s) should be added in the Introduction and for the equations in the paragraph 2.5 Analysis of growth performance and feed utilization parameters;
Response 2: Thank you for your constructive suggestions. We added 4 references in the Introduction (lines 63-68) and paragraph 2.5 (lines 67-68).
Point 3: Regarding Material and Methods, the authors should add more detailed information about rearing conditions like developmental stage (it has never been mentioned if the animals are adult, juvenile, etc.), photoperiod, strain, frequency of feeding, volume of the tanks where the tested animal were distributed and the anesthesia procedures used for the sampling (as mentioned above, the authors should explain why both animals have been raised under the same rearing conditions). Furthermore, the authors should clarify why they followed the procedure of extraction and determination of ASA used in plants in a previous work. The line 143-144 “three weeks later, three replicate independent samples were collected from each experimental group, of which each sample was pooled from five zebrafish or shrimp” is not clear. Finally, in 2.6. Data analysis the authors should mention the post-hoc comparison test used for the ANOVA.
Response 3: We apologize for the unclear presentation of rearing conditions and sample collection information. We rewrote the Material and Methods section in the revised manuscript. The ASA extraction method is also applicable to animal samples, and we have added a reference in Material Methods 2.4. Also, the nutritional content of our basal diets meets the requirements of fish and shrimp growth and development. The feeding environment was suitable, and the fish and shrimp were maintained in good health during the experiment. In addition, we have performed the post-hoc test on the results after one-way ANOVA. The differences among samples were compared using one-way (ANOVA) followed by Tukey’s test in Data analysis.
Point 4: Based on my experience and literature, I have doubts that adult zebrafish can weigh 2 g and even more;
Response 4: We are very apologetic that we did not label it clearly. The weight presented is the total weight of 5 samples, which we have corrected in the revised manuscript.
Point 5: In all the legends to the figures, the statistical analysis used and the number of individuals should be always mentioned. Furthermore, in some figures, results expressed with different units of measurement have been joined in the same graph and this is not correct. It is not necessary to draw on the graph the absence of statistical difference. The description of the results of feed intake (FI) does not match the graph. Finally, the authors should specify that the correlation analysis has been performed for the 2KGA analysis only.
Response 5: We sincerely appreciate the valuable comments. In response to your suggestion, we have made overall modifications to the figure, including the addition of units and significant difference corrections. The correlation analysis in this study focused on the correlation analysis between variation in ASA levels and growth parameters and fertilizer utilization, which contributed us to reveal the mechanism by which 2KGA affects nutrient levels in aquatic animals.
With best regards,
Hui Xu
xuhui@iae.ac.cn

Reviewer 5 Report
Exogenous 2-keto-L-gluconic acid supplementation as a novel approach to enhancing L-ascorbic acid biosynthesis of aquatic animals
Dear Authors,
article is very interesting and describes possibility to increase accumulation of ascorbic acid in aquatic animals, in which plays significant role in case of metabolism by supplementation of 2-keto-L-gluconic acid (2KGA). What important 2KGA increase content of L-ascorbic acid significantly in case aquatic animals treated in conducted experiment and have positive impact for growth rate and health of zebra fish and shrimp. In manuscript there are not much to correct. Below I add some suggestions helpful during this process.
Line 32
In case of units there is lack of central dot: in manuscript is 800 mg kg-1, must be 800 mg·kg-1
Line 49-376
Space is needed before references ‘ …especially in China [1].’
Line 68
In text is: Subsequently, Gao et. al reported… on plant growth [13]’, reference should be added after authors of cited article: ‘Subsequently, Gao et al. [13]…’
Line 103, 104, 120, 124 and 125
The same like in line 32
Line 171
Information about post-hoc test needed in case of one-way ANOVA
Line 180 and 188
The same like in line 32
Line 192
Superscripts: a, b, c could be used to determining statistical significance at p < 0.05; A , B, C at p < 0.01
Line 204
Lack of p-value and superscripts describing differences between treatments
Line 206, 221, 234, 247
Same like in line 180 and 188
Line 258
Maybe it is possible to add also r – value describing strength of correlation ?
Line 269, 271, 352 and 375
The same like in line 32
Line 412-495
Checking of abbreviations is needed, dots on the end of abbreviation,
i.e. in line 412: Environ. Sci. Pollut. Res. Int.
Line 418: Is used name of Journal Reviews in Aquaculture, abbreviation must be used Rev. Aquac.
Line 425: (Tor putitora) must be in italic form
Line 480: Lack of Journal name
Line 495: Change from capital letters to normal form
Please to check the writing convention in information to the authors on the Animals page
Author Response
Replies to Reviewer 5
Dear Reviewer:
We have received your comments on the manuscript entitled: “Exogenous 2-keto-L-gulonic acid supplementation as a novel approach to enhancing L-ascorbic acid biosynthesis of aquatic animals”. Thank you very much for your valuable suggestions and comments! We have made revisions to the manuscript accordingly, and our response is below and marked red.
Point 1: In case of units there is lack of central dot: in manuscript is 800 mg kg-1, must be 800 mg·kg-1
Response 1: Thank you for your suggestion. “mg kg-1” has been replaced by “mg/kg” according the Animals template.
Point 2: Space is needed before references ‘ …especially in China [1].
Response 2: Thank you for your suggestion. Following your comments, we have revised the entire manuscript.
Point 3: Line 68 In text is: Subsequently, Gao et. al reported… on plant growth [13]’, reference should be added after authors of cited article: ‘Subsequently, Gao et al. [13]…’.
Response 3: Thank you for your suggestion. We have revised the sequence. “Subsequently, Gao et al. [17] reported that exogenous 2KGA can increase ASA accumulation and the expression of enzymes related to ASA synthesis and stress resistance, thereby alleviating the inhibitory effects of salt stress on plant growth.” (line 83-86)
Point 4: Line 103, 104, 120, 124 and 125 The same like in line 32.
Response 4: Thank you for your suggestion. “mg kg-1” has been replaced by “mg/kg” according the Animals template.
Point 5: line 171 Information about post-hoc test needed in case of one-way ANOVA.
Response 5: Thank you for your suggestion. We have performed post-hoc test on the results after one-way ANOVA. The differences among samples were compared using one-way (ANOVA) followed by Tukey’s test.
Point 6: Line 180 and 188 The same like in line 32.
Response 6: Thank you for your suggestion. “mg kg-1” has been replaced by “mg/kg” according the Animals template.
Point 7: Line 192 Superscripts: a, b, c could be used to determining statistical significance at p < 0.05; A , B, C at p < 0.01.
Response 7: Thank you for your suggestion. Following your comment, we have revised the presentation of the significance results in the entire manuscript.
Point 8: Line 204 Lack of p-value and superscripts describing differences between treatments.
Response 8: Thanks to your suggestion, we have added statistical information to the revised manuscript.
Point 9: Line 206, 221, 234, 247 Same like in line 180 and 188
Response 9: Thank you for your suggestion. “mg kg-1” has been replaced by “mg/kg” acorroding the Animals template.
Point 10: Line 258 Maybe it is possible to add also r-value describing strength of correlation ?
Response 10: We sincerely appreciate the valuable comments. We have recorrected Figure 8 and added the r-value. (line 302)
Point 11:Line 269, 271, 352 and 375 The same like in line 32.
Response 11: Thank you for your suggestion. “mg kg-1” has been replaced by “mg/kg” according the Animals template.
Point 12: Line 412-495 Checking of abbreviations is needed, dots on the end of abbreviation, i.e. in line 412: Environ. Sci. Pollut. Res. Int.
Response 12: Thanks for your suggestion. “Environ. Sci. Pollut. Res. Int.” has been replaced by “Environ Sci Pollut Res Int.”(line 467)
Point 13: Line 418: Is used name of Journal Reviews in Aquaculture, abbreviation must be used Rev. Aquac.
Response 13: We apologize for our carelessness and thank you for the reminder. “Journal Reviews in Aquaculture” has been replaced by “Rev. Aquac”.(line 559)
Point 14: Line 425: (Tor putitora) must be in italic form.
Response 14: We have made the modification on line 425. “Tor putitora” has been replaced by “Tor putitora”.(line 484)
Point 15: line 480: Lack of Journal name.
Response 15: We have added journal information (line 525).
Point 16: Line 495: Change from capital letters to normal form.
Response 16: Thanks for your suggestion. Following your suggestion, we have modified the references.
Point 17:Please to check the writing convention in information to the authors on the Animals page.
Response 17: Thanks for your suggestion. We have followed the guidance of the authors and have carefully revised it.
With best regards,
Hui Xu
xuhui@iae.ac.cn

Round 2
Reviewer 1 Report
Dear authors,
Author has substantially improved the manuscript and can be accepted for publication.
Reviewer 3 Report
improvements were made to the work according to the requirements of the reviewers, I believe that the work can be published
Reviewer 4 Report
Dear authors,
As requested, I reviewed the revisions in the manuscript “Exogenous 2-keto-L-gluconic acid supplementation as a novel approach to enhancing L-ascorbic acid biosynthesis of aquatic animals” by Meijun Shi, Mingfu Gao, Hao Sun, Weichao Yang, Lixin Zhang and Hui Xu.
Based on the cover letter, the authors have not replied to all of my questions and the paper still shows flaws and weaknesses.
The authors have satisfied the request to improve references in the Introduction and Materials and Methods and they have modified the legends to the figures and the statistical analysis.
However, the crucial point of the paper has not been resolved. The authors keep stating that zebrafish (a teleost) and Pacific white shrimps (arthropoda) have been raised under the same rearing conditions and it is obvious that this is not possible since Pacific white shrimp is a marine animal while zebrafish is not. Furthermore, considering the differences of the two animals, the authors have not explained why both have been administered with the same basal diet (also, they have not stated whether it is a commercial or a laboratory diet). In my opinion, the same basal diet cannot equally meet the nutraceutical requirements of both zebrafish and shrimps. Zebrafish and Pacific white shrimps are two animal models for different reasons, but they cannot be combined for the analysis. The Discussion needs to be more specific. The Results should be discussed keeping in mind the physiology, biochemistry and genetics of zebrafish on one hand and Pacific white shrimp on the other. They cannot be used as general models of aquatic animals.
Based on these considerations, I suggest to reject the paper.
Thank you very much for your attention to my opinion.